# Quantifying LLM Attention-Head Stability: Implications for Circuit Universality

**Karan Bali** [1]   **Jack Stanley** [1]   **Praneet Suresh** [1]   **Danilo Bzdok** [1]

## Abstract

In mechanistic interpretability, recent work scrutinizes transformer "circuits"—sparse, mono or multi layer sub computations, that may reflect human understandable functions. Yet, these network circuits are rarely acid-tested for their stability across different instances of the same deep learning architecture. Without this, it remains unclear whether reported circuits emerge universally across labs or turn out to be idiosyncratic to a particular estimation instance, potentially limiting confidence in safety-critical settings. Here, we systematically study stability across-refits in increasingly complex transformer language models of various sizes. We quantify, layer by layer, how similarly attention heads learn representations across independently initialized training runs. Our rigorous experiments show that (1) middle-layer heads are the least stable yet the most representationally distinct; (2) deeper models exhibit stronger mid-depth divergence; (3) unstable heads in deeper layers become more functionally important than their peers from the same layer; (4) applying weight decay optimization substantially improves attention-head stability across random model initializations; and (5) the residual stream is comparatively stable. Our findings establish the cross-instance robustness of circuits as an essential yet underappreciated prerequisite for scalable oversight, drawing contours around possible white-box monitorability of AI systems.

## 1. Introduction

Modern deep neural networks (DNNs) exhibit a notable phenomenon: DNNs trained at different starting initializations frequently yield solutions with similar prediction performance (Choromanska et al., 2015; Garipov et al., 2018;

Scimeca et al., 2022). Yet, this begs a neglected corollary question today: do current transformers learn radically different sets of intrinsic computational schemes that happen to perform similarly, or do transformers learn internal representations that are largely the same? (Chughtai et al., 2023)

Further, the application scenarios explored in current large language model explainability research (LLM XAI) tend to be limited to narrow tasks and lack generalizability. The extracted "LLM circuits" are rarely validated in unseen settings or tasks (Olah, 2022). Some authors argue (Hendrycks & Hiscott, 2025) that current AI systems are too complex and that a top-down interpretability approach that captures emergent properties is preferable to attempts to reverse-engineer models "neuron by neuron" or in small functional blocks of neurons called "circuits". In other words, by focusing on representations as the primary units of analysis—rather than neurons or small circuits—this approach seeks meaning in patterns of activity across many neurons at a time. Furthermore, existing mechanistic research into transformers often relies on analyzing single attention heads in isolation, effectively cherry-picking learned modules without establishing their representativeness.

These observations can be seen through the lens of the "universality hypothesis" from the mechanistic interpretability literature (Chughtai et al., 2023; Olah et al., 2020). This hypothesis asserts that models learn the same or similar representational features and circuits across different models when trained on similar tasks. However, the strength of the similarities between these learned model components has yet to be systematically quantified, leaving the critical distinction between strong and weak universality unresolved. (Chughtai et al., 2023). Hence, in this work we make the notion of universality experimentally accessible by investigating whether the model converges to the requisite representational space—alignment of the attention head's induced representation, that is, the subspace it attends to and writes into—and therefore to the same functional behavior.

Over the last months and years, we have seen a flurry of advances in LLM circuit discovery methods. These methods involve both automated (Conmy et al., 2023) and manually discovered circuits (Wang et al., 2023; Marks et al., 2025). These methods make sweeping statements about the func-

[1]Mila - Quebec Artificial Intelligence Institute, Montreal, Canada. Correspondence to: Karan Bali <karan.bali@mail.mcgill.ca>.

*Proceedings of the 43rd International Conference on Machine Learning*, Seoul, South Korea. PMLR 306, 2026. Copyright 2026 by the author(s).

tional roles of particular heads in an LLM based on the patterns emerging in their attention score matrices (Elhage et al., 2021; Wang et al., 2023; Olsson et al., 2022; Voita et al., 2019). However, we know that multiple theoretical solutions can exist for the same problem. For example, transformers have been shown to model "backup" circuitry that is utilized only when the primary circuit is compromised. And proxy models for interpretability such as transcoders and sparse autoencoders do not learn canonical concepts, even if they have high reconstruction fidelity and are monosemantic (Olah, 2025).

We face a similar problem in the context of LLM circuits and internal representations in general, where we might have different attention head representations for a retrained model but with different parameter initialization (i.e. a different seed). This hinders the application of LLMs in safety-critical areas like finance, nuclear, or healthcare, where the cost of a single point of failure might be huge (Bengio et al., 2025). We cannot rely on features or circuits extracted from a domain-specialized LLM as explanations of the domain's intrinsic mechanisms, because these structures may be unstable across refits. For example, circuits identified in a model trained on disease-specific data should not be treated as a definitive account of the disease without demonstrated consistency across multiple random initializations of the same model. Without these replications, a seemingly "explainable" feature impacting diagnosis may not have any bearing on the true underlying biological mechanisms, and may in fact be entirely spurious (Stanley et al., 2025).

It is therefore crucial to study the "seed stability" of LLM attention heads, the components most critical in circuits. We ask: if two different teams use the same LLM architecture and the same data, do they end up with the same attention heads? While this may be a straightforward question to articulate, comparing two refits of the same model is not trivial. There exist permutation symmetries of hidden units (Ainsworth et al., 2023); that is, one can swap any two units of a hidden layer in a network and, assuming weights are adjusted accordingly, network functionality will not change. Despite the same or similar output prediction performance, these permutations of parameter spaces can cause various transformations of the internal activation space of complex LLMs, which makes the task of comparing among refits even more challenging (Zhang et al., 2025). Cross-seed comparisons made using methods like CKA, CCA, and SVCCA (Kornblith et al., 2019; Hardoon et al., 2004; Raghu et al., 2017) are not flawless and can be misleading (Kornblith et al., 2019; Davari et al., 2022).

Given this, in the present paper, we compare attention head functions using their attention score matrices, which encode token-token relationships on a common basis and are therefore directly comparable across seeds (Kobayashi et al.,

2024). This choice yields a simple, linear, and easily attributable metric for assessing the seed stability of attention heads.

Our main contributions and findings based on the experiments detailed in Methods (§ 3) and Results (§ 4) sections are the following:

- **Middle-layer instability**: Attention heads in middle transformer layers are the least stable across refits and the most representationally distinct.

- **Depth dependence**: Instability increases with the depth of the transformer model; alternative instances of deeper models exhibit stronger mid-depth divergence across refits.

- **Functional implication of instability**: With increasing depth of the layer, unstable heads become increasingly influential within their respective layer.

- **Effect of weight decay**: Using AdamW (decoupled weight decay) substantially improves attention-head seed stability across model refits.

- **Residual-stream robustness**: Across model refits, the residual stream is relatively much more stable than the attention heads.

From our comprehensive experiments, we discover that the seed stability of a GPT-like architecture is clearly tied to the various choices we make while deciding upon the training configuration of the model (e.g., optimiser, weight decay). These choices as such might have no appreciable impact upon the performance of the final trained model but can have a large impact on the universality of representational structures across model instantiations. This in turn has a cascading effect on the practical usefulness of these representational structures in the broader context of explainability.

## 2. Related Work

Prior studies exploring variation across refits of Transformer models have largely relied on handpicked task subsets or output-level metrics to assess stability. Similar to our work, these works train multiple seeds of transformer-based architectures, but their emphasis is typically on downstream behavior rather than internal alignment. For example, van der Wal et al. (2025) trains multiple Pythia variants across nine seeds and primarily assess downstream performance stability and the representational stability of specific linguistic features via probing. Müller-Eberstein et al. (2023) uses MultiBERTs (five BERT refits)(Sellam et al., 2022) to conduct longitudinal analyses across training checkpoints, focusing on the evolution of an internal task subspace. Similarly, Michaelov et al. (2025) performs a checkpoints-wise

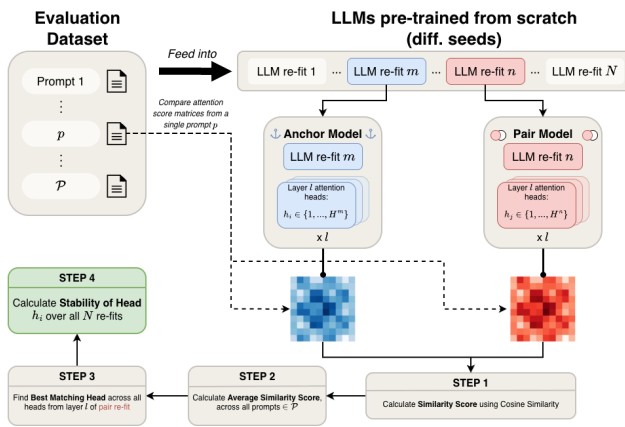

*Figure 1.* Experimental workflow for assessing attention head stability. For each of our 26 architectural configurations, we instantiate $N$ GPT-2-small LLMs, each with a different initial random seed. Each LLM is pre-trained from scratch with the OpenWeb-Text or C4 datasets, depending on the architectural configuration. After pre-training, we take $\mathcal{P}$ evaluation prompts and compute attention score matrices for each head in each layer. An anchor model $m$ is selected and compared to a pair model $n$, recursively for each randomly initialized and pre-trained LLM in that architectural subset. Cosine similarity is used to assess the similarity of each attention score matrix from each head, and a "Stability" score is computed from the best matching heads across refits of the architectural subset. See Methods § 3.2 for definitions of these terms.

longitudinal study over six seeds for three Parc families (Parc-Pythia, Parc-Mamba, Parc-RWKV). Notably, most of these efforts are contingent on architectures other than the decoder-only transformer defined in Vaswani et al. (2017). While Karamcheti et al. (2021) reports training five GPT-family seeds at multiple scales, the paper does not analyze internal representations across refits.

By contrast, our methods performs inter-seed comparison in GPT-like (decoder-only) architectures spanning scales from 2-layer (7M parameters) to 12-layer GPT-2-small (124M parameters; (Radford et al., 2019)). Unlike prior LLM-XAI work, we compare internal representations directly and in a task-agnostic manner by operating on attention score matrices. To the best of our knowledge, no previous work has compared attention heads with their representations across seeds in decoder-only transformers.

## 3. Methods

### 3.1. Architectures and training details

We study attention head similarity in variants of GPT-like (decoder-only) language models. To isolate the intrinsic seed-stability behavior, we hold most training choices fixed at standard defaults, varying only fundamental architectural factors—depth (2, 4, 8) and heads per layer (8 or 16). Following Elhage et al. (2021), we also trained identical

attention-only variants of the above architectures. In addition to this, we also trained a *GPT2-small* architecture with 12 layers and 12 heads configuration.

Our priority was attribution of seed stability in the context of each model's overall architecture. Many hyperparameters such as learning rate schedules, warm up lengths, etc, were kept constant across refits and architectures. This allowed us to isolate and study the seed-stability as a direct outcome of the architecture itself rather than a combination of minor hyperparameter choices.

The only exception to this was the mirror set of identical variants trained with AdamW (Loshchilov & Hutter, 2019) optimizer instead of Adam (Kingma & Ba, 2015), which is motivated by the "norm growth" rationale discussed in Results section 4.4.1. To probe the effect of over- and under-training regimes on seed stability (Hoffmann et al., 2022), we also varied the pretraining corpus and size—the 12 layered GPT2-small models were trained on a bigger 9 billion tokens of OpenWebText (Gokaslan et al., 2019) dataset, whereas the 2, 4, and 8 layered models were trained on a smaller 2 billion token subset of C4 (Nanda, 2022; Raffel et al., 2020) dataset.

In total, we focused on **26 architectures**. We trained 50 refits each for 2, 4, and 8 layered variants and 5 refits each for 12 layered (GPT2-small) architectures. Associated code and complete training/architecture configuration templates are provided in App. A.2, A.3 and the Supplementary Material. Consistent results across these architectural & dataset variations (§ 4) strengthen our analysis & interpretation.

### 3.2. Attention-head stability across refits

We quantify a concrete notion of stability of an attention head by measuring how consistently it can be matched to a head in the **same** layer of independently trained refits of the same architecture. Let there be $N$ refits (same architecture, hyperparameter set and data, yet different random seeds).

SETUP AND NOTATION

- Let $\mathcal{P}$ be a fixed set of prompts.
- Choose an *anchor* refit $m$ and a *pair* refit $n$ ($n \neq m$).
- $h_i$ is a head belonging to layer $l$ in *anchor* refit $m$.
- $h_j$ can be any head belonging to layer $l$ in *pair* refit $n$.
- All possible $h_j$ heads are considered as *candidate heads* to be compared & matched with $h_i$ head.

### 3.2.1. STEP 1: PROMPT-WISE HEAD SIMILARITY

For a head $h_i$ of layer $l$ in anchor refit $m$ and one of its candidate heads $h_j$ in pair refit $n$, we define the prompt-wise similarity between $h_i$ and $h_j$ as the *cosine similarity* between their vectorized (flattened) post-softmax attention score matrices for prompt $p \in \mathcal{P}$, that is $\left(\left(A^m_{h_i}(p)\right), \left(A^n_{h_j}(p)\right)\right)$, from

heads $h_i$ and $h_j$ respectively:

$$s_{(h_i,h_j)}^{(m,n)}(p) = CosineSim\left(\text{vec}\left(A_{h_i}^m(p)\right), \text{vec}\left(A_{h_j}^n(p)\right)\right) \tag{1}$$

### 3.2.2. STEP 2: AVERAGE SIMILARITY SCORE ACROSS PROMPTS

We aggregate over prompts to obtain an average similarity scores between the two heads ($h_i$ and $h_j$):

$$\bar{s}_{(h_i,h_j)}^{(m,n)} = \frac{1}{|\mathcal{P}|} \sum_{p \in \mathcal{P}} s_{(h_i,h_j)}^{(m,n)}(p) \tag{2}$$

### 3.2.3. STEP 3: BEST-MATCH HEAD IN THE PAIR REFIT

By repeating Step 2 (Eq. 2), We compare head $h_i$ against all possible candidate heads $h_j$ in the pair refit ($\{1, \ldots, H^n\}$). Then, we choose the best matching head based on their *average similarity scores*:

$$s_{h_i}^{(m,n)} = \max_{h_j \in \{1, \ldots, H^n\}} \bar{s}_{(h_i,hj)}^{(m,n)} \tag{3}$$

We interpret $s_{h_i}^{(m,n)}$ as the *stability* of head $h_i$ with respect to the single refit $n$. For instance, a representative plot for an 8-layer architecture Fig. A.5 visualizes the stability of all heads from an anchor refit with with respect to other individual pair refits.

### 3.2.4. STEP 4: STABILITY OVER ALL REFITS

Finally, we average the single-refit stability (Eq. 3) across all pair refits to obtain the overall stability of head $h_i$ across all $N$ refits:

$$S_{h_i}^{(m)} = \frac{1}{N-1} \sum_{n \neq m} s_{h_i}^{(m,n)} \tag{4}$$

*Interpretation*: $S_{h_i}^{(m)}$ indicates the propensity of anchor head $h_i$ to be matchable to a head in the *same layer* with a *similar attention-head representation* across independently initialized refits. In this way, we obtain a principled, permutation-invariant notion of attention head stability that obviates the need for Hungarian matching (Ainsworth et al., 2023).

Next, we can repeat this procedure for every head in the anchor refit $m$.

### 3.3. Cross-layer best-match variant

We also consider a relaxation of the § 3.2 procedure that drops the layer constraint when identifying a best match. For an anchor head $(l, h_i)$ in refit $m$, the candidate set includes all heads from all layers (rather than only layer $l$) in the pair refit $n$. That is, $h_j$ can be any head belonging to any layer in *pair* refit $n$. The rest of the method remains identical.

## 4. Results

In this section we present the experimental results that underpin our main contributions. We discuss ten key results that probe cross-seed stability from complementary perspectives. For brevity, we report results only for representative architectures in the main body; the complete set of similar assessments for other model architectures is provided in App. B.

### 4.1. Head-wise and layer-wise stability

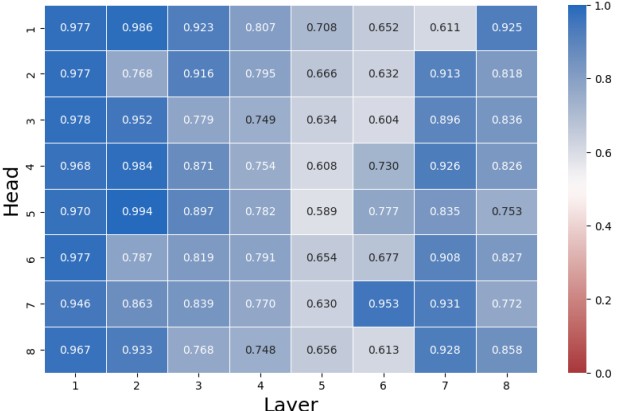

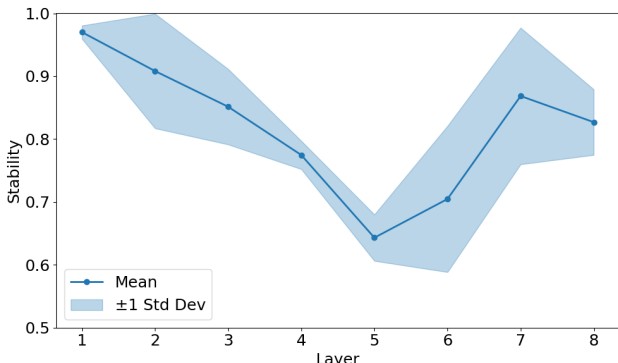

*Figure 2.* **Only some mid-layer attention heads persist across refits.** For each head $h_i$ from layer $l$ in an anchor model, We find its best-matching counterparts in the same layer $l$ across other refits and use $S_{h_i}^{(m)}$ from Eq. 4 as the head's stability (top). This plot thus summarizes inter-seed stability for all the heads of the anchor. Consistently, initial and final layers are more stable than middle layers (bottom).

**Head-wise stability**: Using the metric defined in § 3.2 and a 100-prompts set, we compute the stability of every head in a single **anchor** instance drawn from different refits of a given architecture. The results allow for a bird's eye view of cross-seed stability: many refits are compared against one anchor within the same architecture. For an 8-layer, 8-head MLP architecture (Fig. 2), we consistently observe a higher cross-seed stability in the earlier and later layers, with a pronounced dip in stability measures corresponding to attention heads in the middle layers of the architecture.

At the head level, profiles vary: some layers are uniformly stable (or uniformly unstable) across seeds, whereas others exhibit patterns with variance.

**Layer-wise stability**: From the ensuing head-wise estimates, we compute a layer-wise mean by averaging the stability of all heads within a layer. Here, $h_i \in H$, where $H$ denotes the set of all heads in layer $l$, and $m$ denotes the anchor refit.

$$S_l^{(m)} = \frac{1}{|H|} \sum_{h_i \in H} S_{h_i}^{(m)} \tag{5}$$

Fig. 2 highlights the mid-depth instability. For example, $S_5^{(m)}$ (layer 5) drops to $\approx 0.70$, down from 1.0 in layer 1. This finding highlights the inherent challenge in identifying truly universal circuits in the intermediate layers of transformer models. Comparable figures for other architectures are provided in the § B.1.

## 4.2. Cross-layer best-match stability

**Relaxing the layer constraint**: As noted in § 3.3, we also compute a cross-layer best-match stability by allowing each anchor head to match with any head of any layer in the pair refit. As expected, the resulting stability is slightly higher that the stability calculated with layer constraints (§ 4.1), simply due to the larger candidate pool. However, overall trend was similar. Representative plots for a few of the architectures are provided in the App. B.2.

**Alignment Map & Middle-layer dispersion**: To visualize *"Which layer do the best matches come from?"*, we construct a row-normalized alignment matrix (H) (shown in Fig. 3) with entries $(H_{i,j})$. The y-axis denotes the anchor head's layer index $(i)$; the x-axis denotes the layer index $(j)$ of the best-matched heads. Rows sum to 1. The heatmap exhibits strong diagonal dominance, that is, most heads match to a similar layer in paired refits, confirming a strong layer wise correspondence across seeds.

Compared to early and late layers, middle layers show a broader, flatter row distribution (*lower kurtosis*), with substantial off-diagonal mass into neighboring layers. This cross-layer spillover is also consistent with mid-depth instability and indicates a less sharply defined layer-to-layer correspondence for those heads.

## 4.3. Within-layer uniqueness of attention heads

**Method**: For anchor refit $m$, head $h_i$ in layer $l$ is compared to all other heads in the same layer using attention score matrices over the prompt set $\mathcal{P}$ (as in § 3.2). Additional details are provided in App. A.6. The mean similarity of an attention head to its peers quantifies the commonness (non-uniqueness) of the head.

**Result & Interpretation**: As shown in Fig. 4 and § B.3,

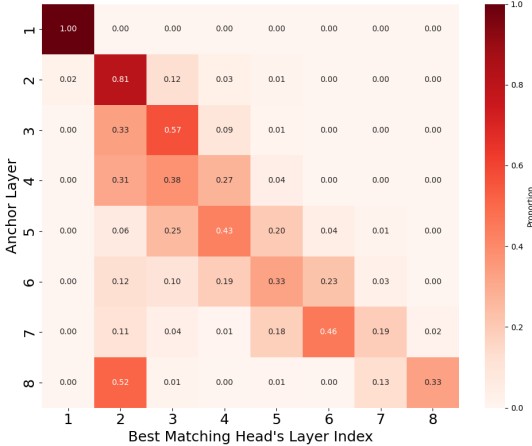

*Figure 3.* **Middle layer attention heads exhibit off-layer matching across re-fits.** Cross-layer best match alignment heatmap shows strong diagonal dominance with heads typically matching to same-depth layers but broader off-diagonal spread in mid-layers.

mid-layer heads are most unique; early/late layers are much more prototypical—mirroring the head stability pattern described in § 4.1.

## 4.4. Factors promoting attention head instability

Building on the layer-wise stability analysis in § 4.1, we sought simple, model-internal correlates of head stability. We focused on parameters that directly shape the attention score matrices, namely query, key, value, and output projection matrices, as well as the pre-attention LayerNorm. Thus, we conducted an exploratory analyses (CKA-based summaries, diagnostic visualizations, and layer-wise statistics) on these objects. Two of the consistent signals that emerged were related to (i) $\ell_2$ norm of the query weights and (ii) prompt length.

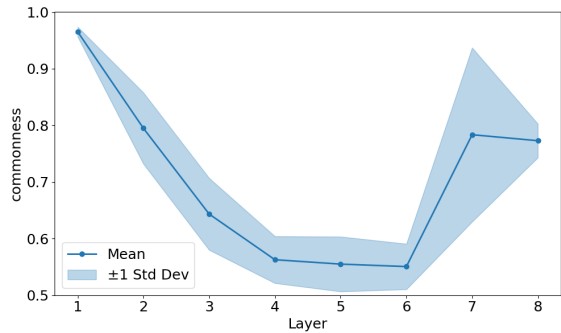

*Figure 4.* **Middle layer attention heads are most unique.** Attention heads are most redundant in early and late layers of GPT-like models, while being quite distinct in the intermediate layers.

### 4.4.1. CORRELATION BETWEEN QUERY-WEIGHT NORM AND LAYER-WISE STABILITY

We compared layer-wise stability from § 4.1 to the mean ($\ell_2$) Frobenius norm of the attention heads' query-weight matrices of the same layer. We observe a clear **negative correlation**—layers with larger mean query-weight norm tend to exhibit lower stability. As shown in Fig. B.5, this trend holds across architectures. This observation is especially intriguing in light of prior work linking parameter ($\ell_2$) norm growth to training instabilities and related failure modes (Lyle et al., 2024; Wortsman et al., 2024). We reproduce these findings in the context of circuit identification and interpretability.

### 4.4.2. EFFECT OF PROMPT LENGTH ON STABILITY

We next recomputed the layer-wise stability using multiple prompt sets whose length spanned 5-50 tokens per prompt. We find that stability decreases sharply with the prompt length, indicating that longer prompts induce greater token-token interaction variability and reduce cross-seed alignment (Fig. 5).

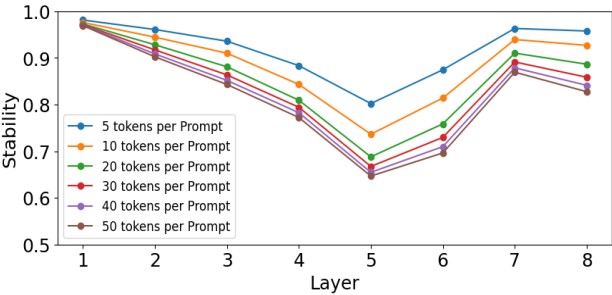

*Figure 5.* Instability of attention heads from middle transformer layers increases with increasing prompt length.

## 4.5. Stability comparison: Adam vs AdamW

**Motivation**: If cross-seed instability partly arises from uncontrolled norm growth, as explored in § 4.4.1, then explicitly regularizing norms during pretraining may provide a means to improve head stability. Prior work shows that AdamW (decoupled weight decay) aids convergence, generalization, and norm control (Loshchilov & Hutter, 2019; Zhang et al., 2018; D'Angelo et al., 2024). Given these findings, it is possible that AdamW also improves the universality of circuit identification across model refits. We therefore trained mirror sets of refits that are architecturally identical to the Adam runs but use AdamW as the optimizer.

**Results**: Studying Figures 6 and App. B.6, we see that AdamW yields a clear increase in head seed-stability for deeper MLP models (8- and 12-layer) and for all attention-only architectures. In contrast, head seed-stability gains for shallow MLP models (2- and 4-layer) are small, possibly

due to limited norm growth at low depth.

**Performance Parity**: Despite the stability gains, replacing Adam with AdamW typically leaves validation perplexity nearly unchanged, indicating that there are no performance tradeoffs when using AdamW. See § B.7 and (Ginsburg et al., 2019).

**Mechanistic check**: To verify that AdamW controls norm growth (Loshchilov & Hutter, 2019), we compared the mean ($\ell_2$) norm of attention-head output activations between matched Adam and AdamW refits. Using AdamW consistently suppresses activation norms, supporting the hypothesis that controlling norm growth improves the cross-seed stability. (See Fig. B.6.1).

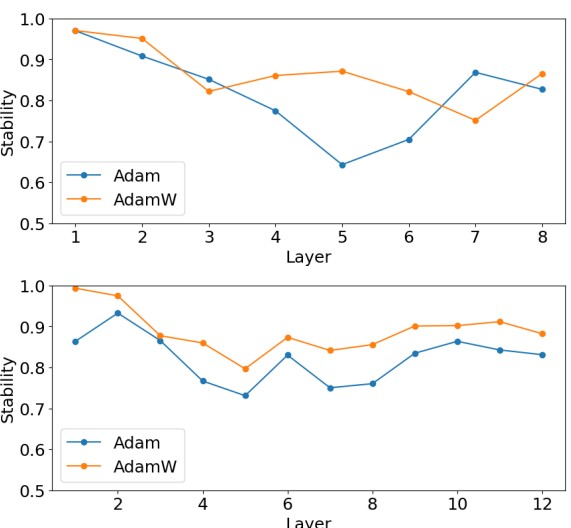

*Figure 6.* AdamW substantially improves attention head seed stability with no significant difference in performance. **Top**: 8-layer, 8-head, MLP architecture. **Bottom**: 12-layer, 12-head, MLP (GPT2-small) architecture.

## 4.6. Most- and least-stable layers

### 4.6.1. STABILITY GAP BETWEEN THE MOST- AND LEAST-STABLE LAYERS

**Definition**: For each of the 26 architectures (§ 3.1), we compare layer-wise stability $S_l$ (§ 4.1) and identify the most-stable layer $l_{\max}$ and least-stable layer $l_{\min}$. We visualize these two layer stability extremities ($S_{l_{\max}}, S_{l_{\min}}$) for an *anchor* refit per model architecture.

To quantify the difference in attention head stability across layers in a single model architecture, we define the stability gap as:

$$\Delta S = S_{l_{\max}} - S_{l_{\min}} \qquad (6)$$

**Results**: (Fig. 7) Across architectures, $\Delta S$ widens with depth, most prominently in 8- and 12-layer models, and is modestly larger for 8-head than for otherwise comparable

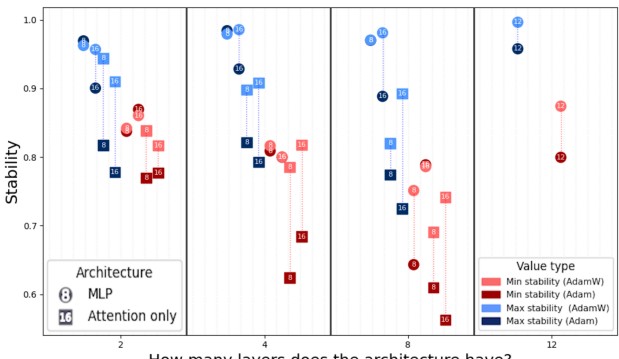

*Figure 7.* $S_{l_{\max}}$ (blue) and $S_{l_{\min}}$ (red) stability vs. depth. Circles: MLP; squares: attention-only. Dark shades: Adam; light: AdamW. Numbers indicate heads per layer. **Architectures trained using AdamW exhibit increased attention head stability.**

16-head models. Using AdamW consistently narrows $\Delta S$ relative to Adam, indicating that decoupled weight decay reduces cross-seed variability within the network itself.

### 4.6.2. POSITION OF THE MOST- AND LEAST-STABLE LAYER RELATIVE TO ARCHITECTURE DEPTH

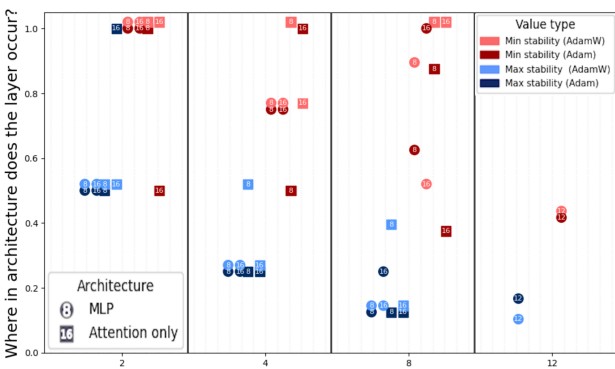

*Figure 8.* $r_{l_{\max}}$ (blue) and $r_{l_{\min}}$ (red) relative depth vs. total layers. Circles: MLP; squares: attention-only. Dark shades: Adam; light: AdamW. Numbers indicate heads per layer. $l_{\min}$ **tends to occur after $l_{\max}$ across architectures and, for 8- and 12-layer architectures, it occurs around the mid-depth.**

**Definition**: For each of the 26 architectures (§ 3.1), we locate the most-stable layer ($l_{\max}$) and the least-stable layer ($l_{\min}$) from the layer-wise stability curve for an anchor refit (§ 4.1). We calculate their relative depth ($r_l$). Here, $l$ is the layer index in $(1, ., l, .., L)$ and $L$ is the total number of layers in the model.

$$r_l = l/L \qquad (7)$$

**Results**: (Fig. 8) Across nearly all configurations—aside from the inherently shallow 2-layer cases—**the least-stable layer occurs after the most-stable layer** and, in 8- and 12-layer models, **occurs around mid-depth** ($r \approx$ in range $[0.4, 0.8]$), providing evidence that instability peaks in the

middle of the network. For example, in the 8-layer, 8-head MLP (Adam) anchor refit (Fig. 8), the most-stable layer is ($\ell = 1$ with $r = 0.125$) while the least stable layer is ($\ell = 5$ with $r = 0.625$). Optimizer choice (Adam vs. AdamW) has no significant effect on these positions.

### 4.7. Stability of residual stream

**Motivation**: Attention heads exhibit notable cross-seed variability (§ 4.1), especially at mid-depth, yet refits still produce similar functional behavior. This suggest that downstream representations in the model's residual stream may be more stable than the individual head representations that feed into it, providing more stable ground for possible universal circuits discovery.

**Method**: Using a set of 100 prompts $\mathcal{P}$, we compute layer-wise residual-stream stability between an anchor refit and a pair refit. For each layer, we use a set of residual stream activations after the attention sub layer is added to the residual stream (i.e. post-attention residual addition). We use these activations to compare anchor and pair refits by utilizing Centered Kernel Alignment (CKA). For CKA, we form centered Gram matrices for each representation set and compute their normalized similarity (Kornblith et al., 2019; Gretton et al., 2005). CKA is invariant to isotropic rescaling and orthogonal transforms, but not to arbitrary affine or nonlinear transforms; thus it can underestimate similarity when refits differ by transformations outside the scope of CKA. Further details about the methodology used for this result are provided in App. A.7.

**Results**: Across architectures and optimizers, the residual stream is consistently more stable than the corresponding attention heads (Fig. 9; B.8; Kornblith et al., 2019). Optimizer choice has no major effect on residual-stream stability.

**Interpretation**: Despite head-level divergence, the objective and shared residual pathway (via skip connections) might nudge the residual stream toward seed-robust representations, helping explain *"why independent refits yield similar outputs despite differing attention patterns"*; consistent with (Xiong et al., 2020; Takase et al., 2023; Wang et al., 2022).

### 4.8. Layer-wise correlation between the stability and post-ablation change in perplexity

**Method**: For each layer $l$, we compute the Pearson correlation between head stability ($S_{h_i}$) and functional importance, quantified by post-ablation change in perplexity ($\Delta \text{PPL}_{h_i}$); additional details are provided in App. A.8.

**Results**: Although per-layer correlations are small and somewhat noisy, the aggregate trend is negative across layers (Figure 10 and B.9).

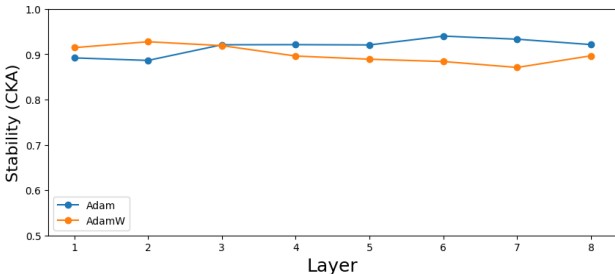

*Figure 9.* The transformer residual stream is relatively much more stable than corresponding attention head that feed into it.

**Interpretation**: These results suggest unstable heads are much more important in later layers for model performance than unstable heads in the earlier layers.

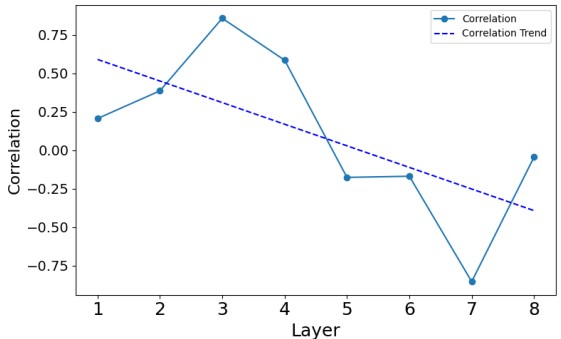

*Figure 10.* Unstable heads become increasingly more important with the depth of the layer.

### 4.9. Geometric observation of attention head activations using meta-SNE

**Objective**: We sought a geometry-aware view of attention heads across architectures and refits by embedding their activation patterns into a shared space using meta-SNE (Olah, 2015). Each point denotes one head from one refit, across architectures; see App. A.9.

**Analysis**: (Fig. 11) When colored by relative depth (See 4.6.2), we observe a clear pattern where embedded heads are clustered by relative depth. This ordering reveals that the functional role of an attention head is determined more by its relative position in the computational stack than by specific architectural configurations (e.g. absolute number of layers).

## 5. Conclusion

Our comprehensive evaluation of the "seed stability" of transformer architectures draws a new boundary for mechanistic interpretability: the internal computational schemes (circuits) of LLMs are not as universal as their output pre-

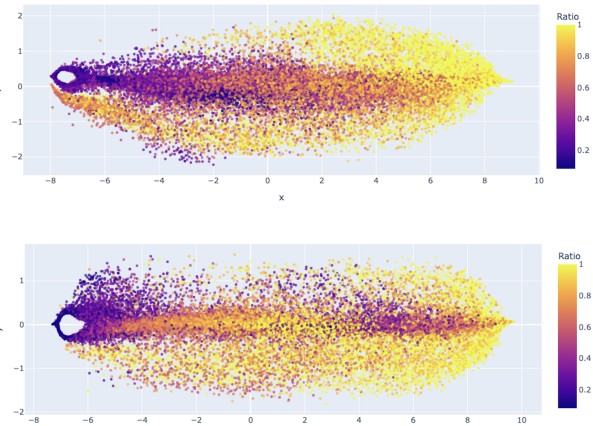

*Figure 11.* Visualizing each attention head's activation space using meta-SNE, when colored by relative depth. Heads are clustered by their relative position in the computational stack of the transformer model. **Top**: Adam-based architectures. **Bottom**: AdamW-based architectures.

diction performance may suggest. We here demonstrate that structural universality of attention heads is a matter of continuous degree that varies by computational depth, optimizer choice during pre-training, and layer-wise uniqueness.

In particular, our results reveal a "stability dip" in middle layers of transformer architectures, where attention heads are the least stable across model refits, and also most unique. Paradoxically, we find that these unstable heads in deeper layers become more functionally influential, suggesting that the most critical components for model performance may be the hardest to replicate across instances. Still, there is hope for the isolation of more universal circuits in transformer models. First, the residual stream demonstrates higher cross-seed stability than individual attention heads, pointing towards the shared residual pathway as an important consolidator of universal interpretable functional units. Second, we find that the use of AdamW solvers substantially improves attention head stability with no model performance penalty. Simple adjustments to model training can hence have outsized impact on improving the universality extracted interpretable components from transformers.

This observed "stability dip" in the middle layers is likely because these intermediate computations operate with the highest degrees of freedom, furthest from the regularizing anchors of the input embeddings and output logits. Without these constraints, individual refits rely on unrestricted parameter expansion, as evidenced by query-weight norm growth shown in this work, to solve tasks via idiosyncratic, seed-specific attention heads. In practical terms at the single-instance level, middle-layer instability aligns with the structural emergence of an intermediate "compression valley" characterized by high representational divergence from the

input space (Skean et al., 2025), and has been linked to the activation of coherent but input-insensitive semantic concepts that may drive hallucinations (Suresh et al., 2025).

By introducing explicit optimization constraints such as coupled weight decay (AdamW), we restrict the parameter space, forcing intermediate layers to converge on more canonical, stable, and universal structures while retaining their task performance. Through an alternate mechanism, the residual stream acts as an architectural constraint, consolidating highly variable head-level outputs into a stable functional pathway. Ultimately, for mechanistic interpretability to scale reliably, circuit extraction must account for this framework, prioritizing more highly constrained regions of the network or enforcing regularization during pre-training to provide stronger guarantees for circuit robustness and reliability.

In particular, we believe this work offers useful guidance for researchers in mechanistic interpretability and related areas, especially those studying or optimizing the attention block for attention-specific interventions such as steering methods or distillation objectives based on divergences between attention patterns.

Without establishing robustness across model instances, extracted LLM "circuits" risk representing stochastic artifacts rather than robust accounts of a model's intrinsic mechanisms. Our findings open the door for more principled monitoring techniques, urging the community to move beyond single-instance analysis of transformer models and toward a more rigorous, stability-tested framework for AI safety.

**Limitations:** Our focus was to study and understand the mechanisms underlying trends in seed stability in LLMs. At the outset, we did not attempt an exhaustive hyperparameter sweep. The only exception was the inclusion of optimizer variants (Adam vs AdamW) for the reasons noted above. However, a study exploring and analyzing the relation between seed stability and a more comprehensive set of hyperparameters is left for future work. Budget and resource considerations also guided our choice of model sizes and variants.

## Acknowledgments

We gratefully acknowledge support from the Canadian Institute for Advanced Research (**CIFAR**), the National Institutes of Health (**NIH**), the Canada First Research Excellence Fund (**CFREF**), and **Coefficient Giving**, formerly Open Philanthropy.

## Impact Statement

Our work has direct bearing for isolating and characterizing universal circuits in transformers and LLMs. These results may yield positive societal outcomes through an enhanced ability to build safe and human understandable AI. Conversely, as for any fundamental mechanistic interpretability research, it is possible that this research may be used to design more sophisticated jailbreak or hijacking attacks on LLMs, or may be used to evade LLM alignment. On net, we expect this interpretability work to have a positive impact for society.

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

## Appendix Index

## A. Methods

### A.1. Datasets

We train on two corpora:

- **C4 (2B token subset)**: All models with 2, 4, or 8 layers are pretrained on a 2-billion token subset of C4. The dataset is available at (Nanda, 2022)
- **OpenWebText (9B tokens)**: All 12-layer models are pretrained on 9 billion tokens on OpenWebText, an open-source replication of OpenAI's WebText used for GPT-2. The dataset is available at (Gokaslan et al., 2019)

### A.2. Architecture details

All architectures are decoder-only Transformers. Each architecture is trained across multiple independent seeds (refits). We consider both **with-MLP** and **attention-only** variants. We used TransformerLens API (Nanda & Bloom, 2022) to instantiate the models. Architecture configuration card templates for variants with a given number of layer (NUM_LAYERS) and number of heads per layer (NUM_HEADS) are provided below (Listing 1 and 2). GPT2-small template has mostly similar configuration with a different tokenizer (Figure 3).

In total we trained 26 architectures. The architectural details are provide in the Table 1. We trained 50 independently initialized refits (random seeds) for every architectures, except GPT2-small types, for which we trained 5.

The supplementary material, related code and model weights for trained refits, is provided in the repositories listed below.

- **Code:** `https://github.com/karanbali/attention_head_seed_stability`
- **Weights:** `https://huggingface.co/karanbali/attention_head_seed_stability`

| # | Layers | Heads | Attention-only | Optimizer | Note |
|---|--------|-------|----------------|-----------|------|
| 1 | 2 | 8 | True | Adam | |
| 2 | 2 | 8 | True | AdamW | |
| 3 | 2 | 8 | False | Adam | |
| 4 | 2 | 8 | False | AdamW | |
| 5 | 2 | 16 | True | Adam | |
| 6 | 2 | 16 | True | AdamW | |
| 7 | 2 | 16 | False | Adam | |
| 8 | 2 | 16 | False | AdamW | |
| 9 | 4 | 8 | True | Adam | |
| 10 | 4 | 8 | True | AdamW | |
| 11 | 4 | 8 | False | Adam | |
| 12 | 4 | 8 | False | AdamW | |
| 13 | 4 | 16 | True | Adam | |
| 14 | 4 | 16 | True | AdamW | |
| 15 | 4 | 16 | False | Adam | |
| 16 | 4 | 16 | False | AdamW | |
| 17 | 8 | 8 | True | Adam | |
| 18 | 8 | 8 | True | AdamW | |
| 19 | 8 | 8 | False | Adam | |
| 20 | 8 | 8 | False | AdamW | |
| 21 | 8 | 16 | True | Adam | |
| 22 | 8 | 16 | True | AdamW | |
| 23 | 8 | 16 | False | Adam | |
| 24 | 8 | 16 | False | AdamW | |
| 25 | 12 | 12 | False | Adam | * GPT2-small with Adam |
| 26 | 12 | 12 | False | AdamW | * GPT2-small with AdamW |

*Table 1.* Details of 26 architectures

## A.3. Training Configuration

Unless noted otherwise, training choices are held at standard defaults to isolate architectural effects; **Adam** (constant learning rate) is the baseline optimizer, with a mirror set trained using **AdamW** (decoupled weight decay). Training configuration templates are shown in (Listing 4, 5, 6 and 7). We define these templates using **TransformerLens HookedTransformer-TrainConfig**; any hyperparameters not listed use the default values. More details can be found in training code provided in supplementary material.

## A.4. Prompts Sets & Generation

We used two prompts resources:

- **Primary set (100 prompts)**: Used for all experiments except § 4.4.2. This set was synthesized using ChatGPT (GPT-5.2 Thinking) model; provided at § A.4.
- **Length-sweep set (6 x 20 prompts)**: Six subsets derived from the same 20 base prompts by truncation, yielding target length of 5, 10, 20, 30, 40, and 50 tokens per prompt. This set was used for § 4.4.2 and is shown in § A.4.

Actual prompt sets are provided within supplementary material.

```
Instruction prompt to generate Primary set of 100 prompts

You are generating an evaluation set for a **non{instruction-tuned, GPT-2-small{style**
language model.  Produce **exactly 100** diverse **text prompts (prefixes)** that a
model can naturally continue.
CRITICAL OUTPUT REQUIREMENTS (to avoid Python SyntaxError):  – Output format:  **a
single valid Python list literal** of **exactly 100** elements, e.g.  ["...", "...",
```

...] – Use **ONLY double-quoted Python strings** for every prompt. – Do **NOT** use triple quotes (no """ and no '''). – Do **NOT** include any unescaped double quotes inside strings. If a prompt would include quotes, use **single quotes** in the text (') or escape as " Escape special characters when needed: backslash as
and newlines as
n. – Do **NOT** wrap the output in markdown fences, and do **NOT** include any explanation text before or after the list.
PROMPT CONTENT REQUIREMENTS: – Each prompt is **15{80 tokens** (roughly 1{4 sentences) and is an **incomplete prefix** (it should not feel finished). – Prompts must be **random, varied, and non-duplicative** (no near-copies). – Avoid requiring external knowledge; questions should be answerable from context or be \continue the story" type. – Keep content safe: no hate, sexual content, self-harm, instructions for wrongdoing, or personal data.
DIVERSITY REQUIREMENTS (spread across the 100 prompts): – Genres: realistic, fantasy, sci-fi, mystery, romance (PG), humor, horror (mild), slice-of-life. – Formats: narration, dialogue, interview transcript, news-style paragraph, diary entry, recipe step-by-step, meeting notes, customer support chat, legal/contract-ish clause, scientific abstract-ish intro. – Structured text: bullet list starts, numbered list starts, parentheses/brackets, quoted speech using single quotes, markdown-like headers, a short poem start, a letter/email opening, a \Q: ... A:" snippet. – Lightweight reasoning: simple arithmetic embedded in text, basic logic constraints, pattern completion (e.g., \A, B, C, ..."), cause-effect setups. – Code-ish prefixes: include a few prompts that begin a function or config snippet (Python/JSON/pseudocode) but do not require correctness|just naturally continuable. Ensure all code-like lines are inside the same double-quoted string and use
n for line breaks.
FINAL CHECK BEFORE YOU OUTPUT: – Count elements: exactly 100 strings. – Ensure the entire output can be pasted into Python without errors.
Now output only the Python list literal of 100 prompt strings.

## Instruction prompt to generate "Length-sweep sets" with varying prompt length

You are generating an evaluation set for a **non{instruction-tuned, GPT-2-small{style** language model.
GOAL: Create **6 length-controlled sets** of prompts by sweeping the hyperparameter **i ∈ [5, 10, 20, 30, 40, 50]**. Each set contains **exactly 20 prompts**. The prompts should be **random, varied, and naturally continuable** prefixes.
CRITICAL LENGTH RULE (how to count \tokens"): – Define \token" as a **space-separated word** using this exact rule: `tokens = text.split(" ")`. – Therefore, each prompt must contain **exactly i tokens** meaning **exactly i space-separated chunks**. – Use **single spaces only** between tokens. No leading/trailing spaces.
CONSISTENCY RULE ACROSS SETS: – The 20 prompts represent 20 \base prompts" (same identity across lengths). – For each base prompt k (1..20), create a **master version of exactly 50 tokens**. – Then derive shorter versions by **truncation**: – The i=5 version is the **first 5 tokens** of the master. – The i=10 version is the **first 10 tokens** of the master. – Similarly for i=20,30,40,50. – This ensures: **only the number of tokens changes**, and content is otherwise identical across sets (shorter is a prefix of longer).
CONTENT REQUIREMENTS (for the 20 master prompts): – Each master prompt must be **50 tokens exactly** (per the split rule). – Prompts must be **incomplete prefixes** (do not feel finished; okay to end mid-sentence). – Prompts must be **diverse** across the 20 items: – Genres/voices: realistic, fantasy, sci-fi, mystery, mild horror, humor, slice-of-life. – Formats: dialogue, diary entry, news-style lead, meeting notes, customer support chat, recipe steps, faux scientific abstract opening, legal-ish clause opening, Q/A snippet, poetic start. – Include a few with lightweight structure: a short numbered list or bullet-like tokens, parentheses, brackets. – Include a few with lightweight reasoning cues (simple arithmetic or constraints) but not requiring correctness. – Include 2{3 code-ish natural prefixes (e.g., \def", \", \if") BUT still plain text; no newlines.
SAFETY REQUIREMENTS: – No hate, sexual content, self-harm, personal data, or instructions for wrongdoing.
OUTPUT FORMAT REQUIREMENTS (to avoid Python SyntaxError): – Output **ONLY** a single valid **Python dict literal** with exactly these integer keys: 5, 10, 20, 30, 40, 50 – Each key maps to a Python list of **exactly 20** strings. – Use **ONLY double-quoted

```
Python strings** for every prompt.  - Do **NOT** use triple quotes (no """ and no ''').
- Do **NOT** include any unescaped double quotes inside strings (avoid double-quote
characters entirely).  - No newlines inside strings.  - Do not add comments, markdown,
or any text before/after the dict.
FINAL CHECK BEFORE OUTPUT: - For every i, every string has exactly i tokens by the
split rule.  - For each k in 1..20, dict[5][k] equals the first 5 tokens of dict[50][k],
dict[10][k] equals first 10 tokens of dict[50][k], etc.  - Exactly 6 keys and exactly 20
prompts per key.
Now output the Python dict literal.
```

## A.5. Stability of head $h_i$ w.r.t other individual pair refit

Figure 12 visualizes per-head stability with respect to 50 individual pair refits for an 8-layer, 8-head MLP model. It shows a series of heatmaps, one per layer, for all heads in the anchor refit (seed = 1). The x-axis indexes pair refits, so each cell reports the stability of a single head against a single pair refit. See Figure 12.

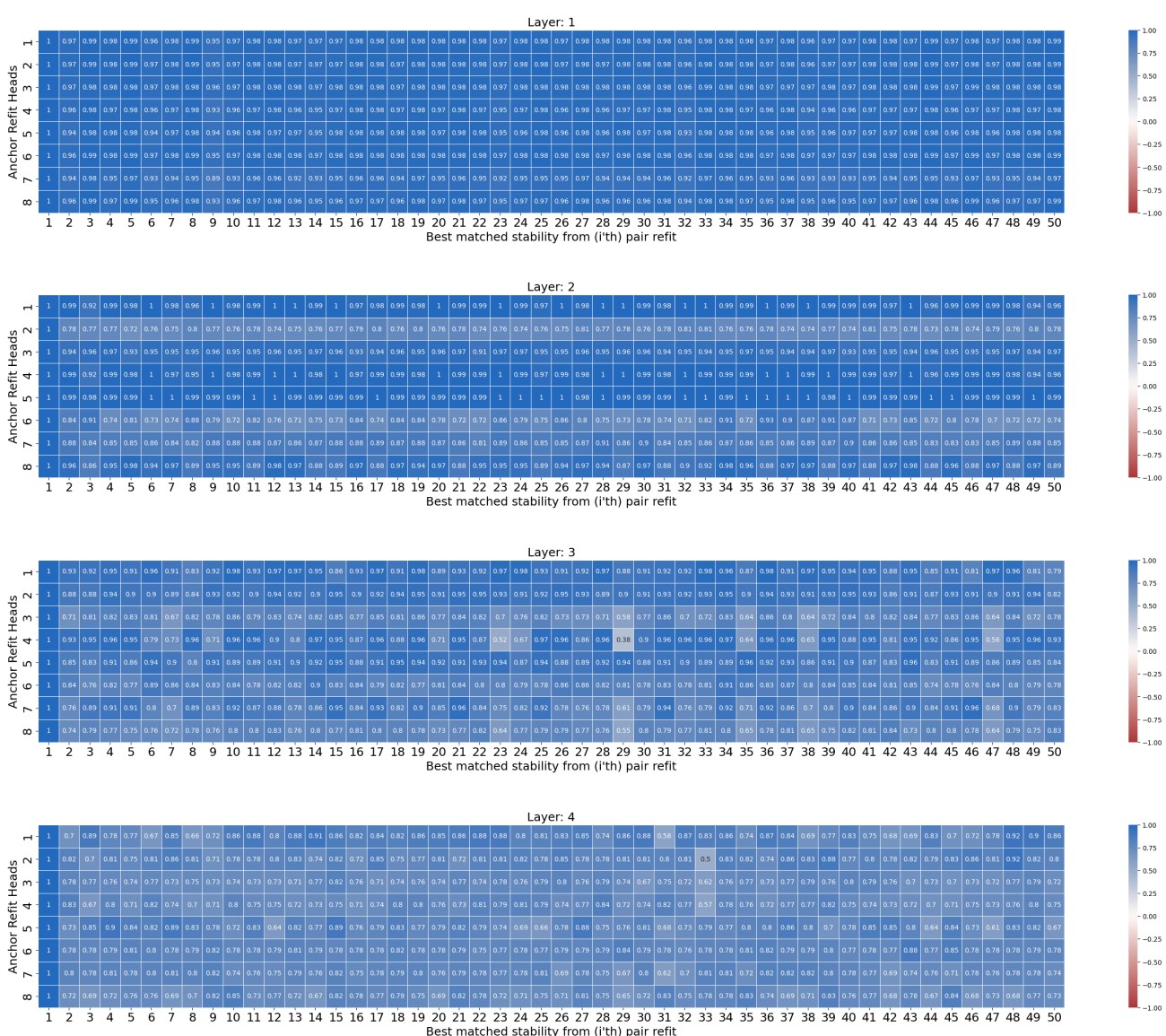

*Figure 12.* Residual stream is relatively more stable than its corresponding attention heads.

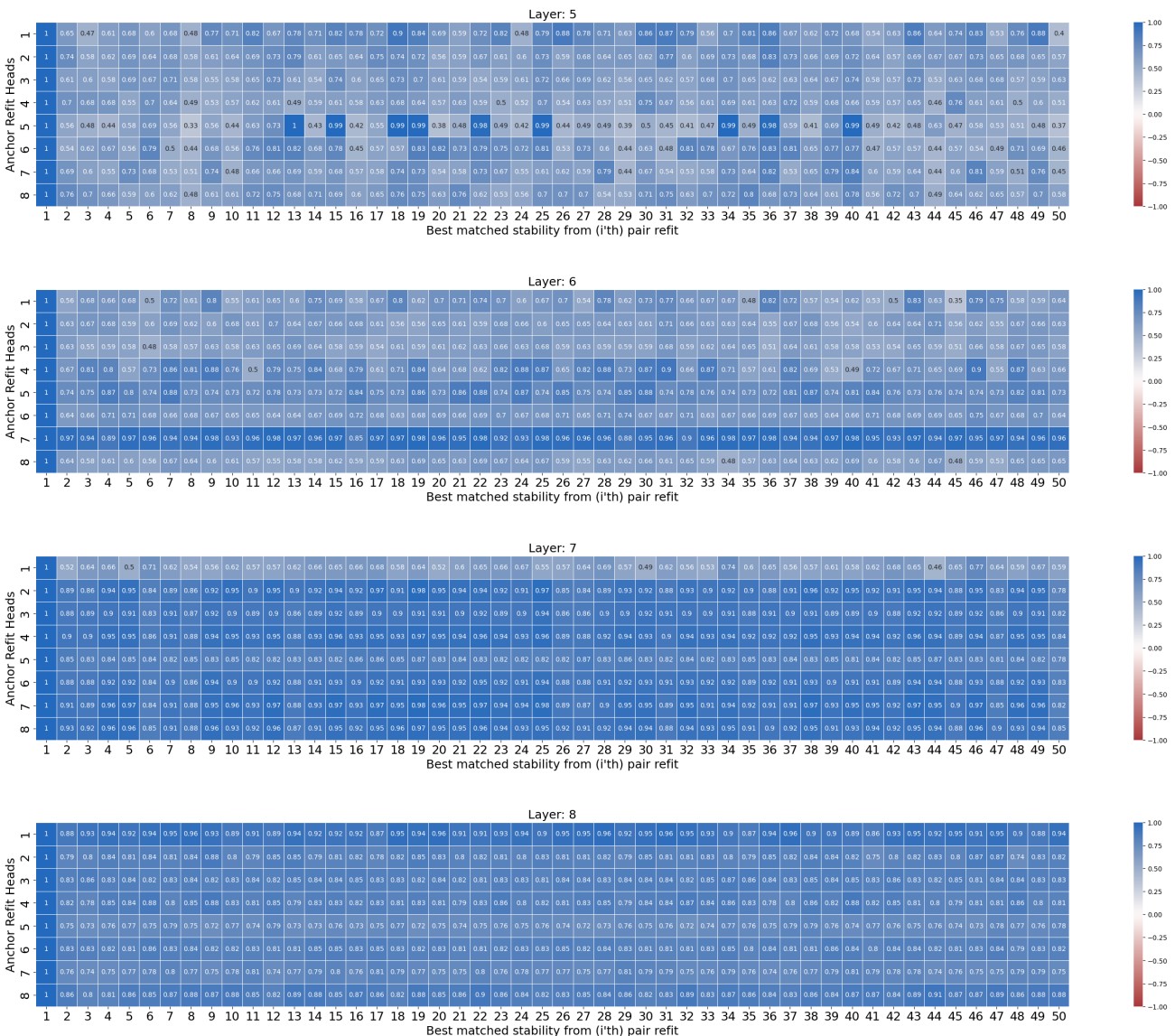

*Figure 12.* Residual stream is relatively more stable than its corresponding attention heads.

## A.6. Within-layer uniqueness of attention heads

Methodology to calculate commonness (non-uniqueness) of an attention head $h_i$ (See § 4.3):

SETUP AND NOTATION

- Let $\mathcal{P}$ be a fixed set of 100 prompts.
- $h_i$ is a head belonging to layer $l$ in a refit $m$.
- $h_j$ can be any head belonging to layer $l$ in the same refit $m$.
- All possible $h_j$ heads are considered as *peer heads* to be compared with.

A.6.1. STEP 1: PROMPT-WISE HEAD SIMILARITY

For a head $h_i$ of layer $l$ in anchor refit $m$ and one of its peer heads $h_j$, we define the prompt-wise similarity between $h_i$ and $h_j$ as the *cosine similarity* between their vectorized (flattened) attention score matrices for prompt $p \in \mathcal{P}$, that is

$\left(\left(A^m_{h_i}(p)\right),\left(A^m_{h_j}(p)\right)\right)$, from heads $h_i$ and $h_j$ respectively:

$$C^{(m)}_{(h_i,h_j)}(p) = CosineSim\left(\text{vec}\left(A^m_{h_i}(p)\right), \text{vec}\left(A^m_{h_j}(p)\right)\right) \tag{8}$$

A.6.2. STEP 2: AVERAGE SIMILARITY SCORE ACROSS PROMPTS

We aggregate over prompts to obtain an average similarity scores between the two heads ($h_i$ and $h_j$):

$$\bar{C}^{(m)}_{(h_i,h_j)} = \frac{1}{|\mathcal{P}|} \sum_{p \in \mathcal{P}} C^{(m)}_{(h_i,h_j)}(p) \tag{9}$$

A.6.3. STEP 3: COMMONNESS OF A HEAD

By repeating Step 2 (Eq. 9), We compare head $h_i$ against each peer heads $h_j \in \{1, \ldots, H^m\}$. Averaging these head-to-head comparisons yields per-head redundancy score, quantifying how common (or unique) $h_i$ is within its layer:

$$Commonness^{(m)}_{(h_i)} = \frac{1}{Number\ of\ heads} \sum_{h_j \in \{1,\ldots,H^m\}} \bar{C}^{(m)}_{(h_i,h_j)} \tag{10}$$

A.6.4. STEP 4: AVERAGE REDUNDANCY OF A LAYER

The average of calculated commonness from Step 3 (Eq. 10) for all the heads $h_i \in \{1, \ldots, H^m\}$ provides a sense of average redundancy within the layer:

$$Commonness^{(m)}_l = \frac{1}{Number\ of\ heads} \sum_{h_j \in \{1,\ldots,H^m\}} \overline{Commonness}^{(m)}_{(h_i)} \tag{11}$$

We can now plot average redundancy for all layers of the architecture at hand, which shows that middle layer attention heads are more unique, relative to those in early and later layers (See Fig. 4 and § B.3)

### A.7. Stability of residual stream

We followed the CKA implementation provided in (Kornblith et al., 2019) to compare layer counterparts of different refits using their set of activations over the prompt set $\mathcal{P}$. To compute centered Gram matrices we used an RBF kernel with a threshold of 1.0.

### A.8. Layer-wise correlation between the stability and "post-ablation change in perplexity"

To probe implications of head instability, we conduct a head ablation study on the anchor refit. For each layer $l$, we:

1. For all the heads $h_i \in l$, we calculate the head's stability ($S_{h_i}$).

2. For all the heads $h_i \in l$, we measure the post-ablation change in perplexity.

We define the "post-ablation change in perplexity" as change in perplexity after ablating the head $h_i$, that is, zero'ing the head $h_i$ output during the forward pass. The perplexity was calculated based on a 100-prompts set § A.1:

$$\Delta PPL_{h_i} = PPL_{ablated}(h_i) - PPL_{baseline} \tag{12}$$

For each layer, we then compute the correlation between two sets containing the:

1. $\{S_{h_i}\}^H_{h_i=1}$ : Stability for all the heads $h_i \in$ layer $l$

2. $\{\Delta PPL_{h_i}\}^H_{h_i=1}$ : "post-ablation change in perplexity" for all the heads $h_i \in$ layer $l$

## A.9. Geometric observation of the attention head activations using meta-SNE

**Method for calculating meta-SNE embeddings**:

Following the approach laid out in (Olah, 2015), for each head $h$, we represent its behavior by using its output activations over a fixed prompt set $\mathcal{P}$. We compute a pairwise distance matrix $D$ whose $D_{i,j}$ entry reflect the dissimilarity between pairs of points within the representation space of head $h$. The points are attention head $h$ output activation for different prompts in $\mathcal{P}$. We use Euclidean distance to calculate the distance matrix $D$. Because meta-SNE consumes distances rather than raw coordinates, it is insensitive to many isometric transformations (common in representation spaces), preserving both local and global neighborhood structure. The resulting distance matrix is then embedded with t-SNE to produce a 2-D embeddings.

Related figures are provided in § 4.9.

# B. Results

## B.1. layer-wise stability

Additional architecture-wise plots for § 4.1:

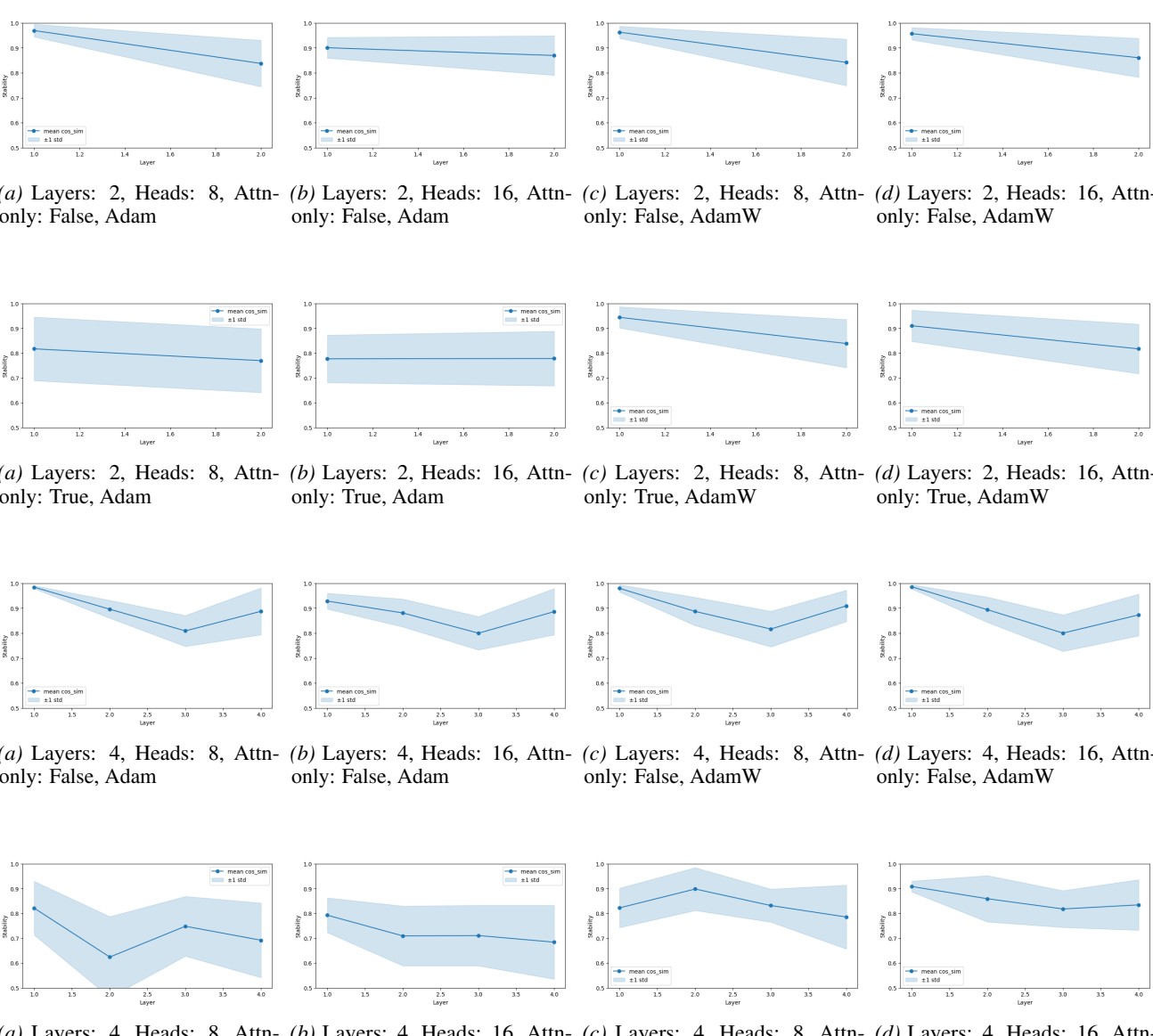

*(a)* Layers: 2, Heads: 8, Attn-only: False, Adam
*(b)* Layers: 2, Heads: 16, Attn-only: False, Adam
*(c)* Layers: 2, Heads: 8, Attn-only: False, AdamW
*(d)* Layers: 2, Heads: 16, Attn-only: False, AdamW

*(a)* Layers: 2, Heads: 8, Attn-only: True, Adam
*(b)* Layers: 2, Heads: 16, Attn-only: True, Adam
*(c)* Layers: 2, Heads: 8, Attn-only: True, AdamW
*(d)* Layers: 2, Heads: 16, Attn-only: True, AdamW

*(a)* Layers: 4, Heads: 8, Attn-only: False, Adam
*(b)* Layers: 4, Heads: 16, Attn-only: False, Adam
*(c)* Layers: 4, Heads: 8, Attn-only: False, AdamW
*(d)* Layers: 4, Heads: 16, Attn-only: False, AdamW

*(a)* Layers: 4, Heads: 8, Attn-only: True, Adam
*(b)* Layers: 4, Heads: 16, Attn-only: True, Adam
*(c)* Layers: 4, Heads: 8, Attn-only: True, AdamW
*(d)* Layers: 4, Heads: 16, Attn-only: True, AdamW

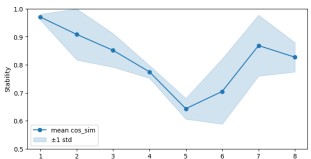 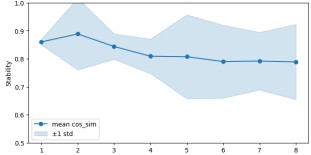 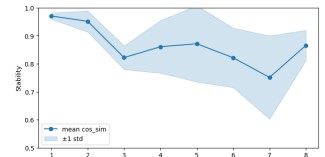 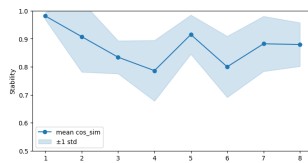

*(a)* Layers: 8, Heads: 8, Attn-only: False, Adam

*(b)* Layers: 8, Heads: 16, Attn-only: False, Adam

*(c)* Layers: 8, Heads: 8, Attn-only: False, AdamW

*(d)* Layers: 8, Heads: 16, Attn-only: False, AdamW

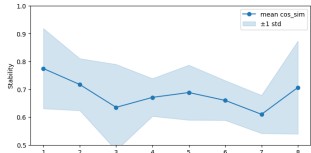 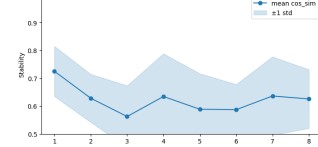 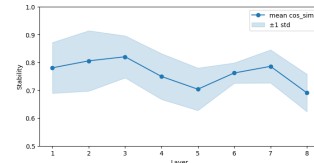 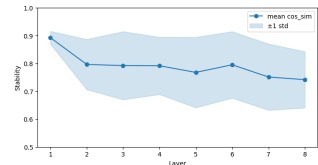

*(a)* Layers: 8, Heads: 8, Attn-only: True, Adam

*(b)* Layers: 8, Heads: 16, Attn-only: True, Adam

*(c)* Layers: 8, Heads: 8, Attn-only: True, AdamW

*(d)* Layers: 8, Heads: 16, Attn-only: True, AdamW

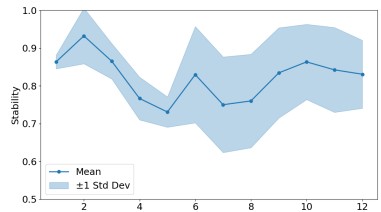 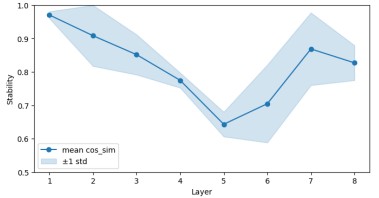

*(a)* Layers: 12, Heads: 12, Attn-only: False, Adam

*(b)* Layers: 12, Heads: 12, Attn-only: False, AdamW

## B.2. Cross-layer best-match stability

Representative plots for few of the architectures § 4.2:

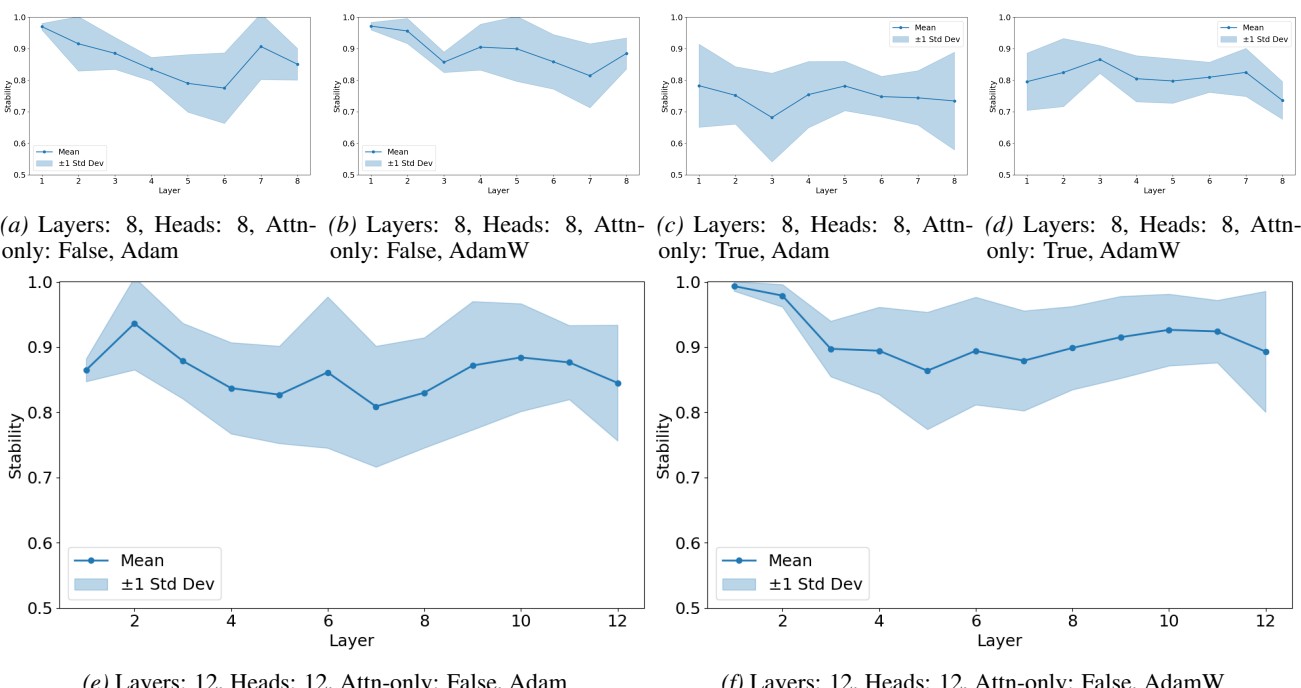

*(a)* Layers: 8, Heads: 8, Attn-only: False, Adam

*(b)* Layers: 8, Heads: 8, Attn-only: False, AdamW

*(c)* Layers: 8, Heads: 8, Attn-only: True, Adam

*(d)* Layers: 8, Heads: 8, Attn-only: True, AdamW

*(e)* Layers: 12, Heads: 12, Attn-only: False, Adam

*(f)* Layers: 12, Heads: 12, Attn-only: False, AdamW

*Figure 20.* Representative plots for cross-layer best-match stability § 4.2

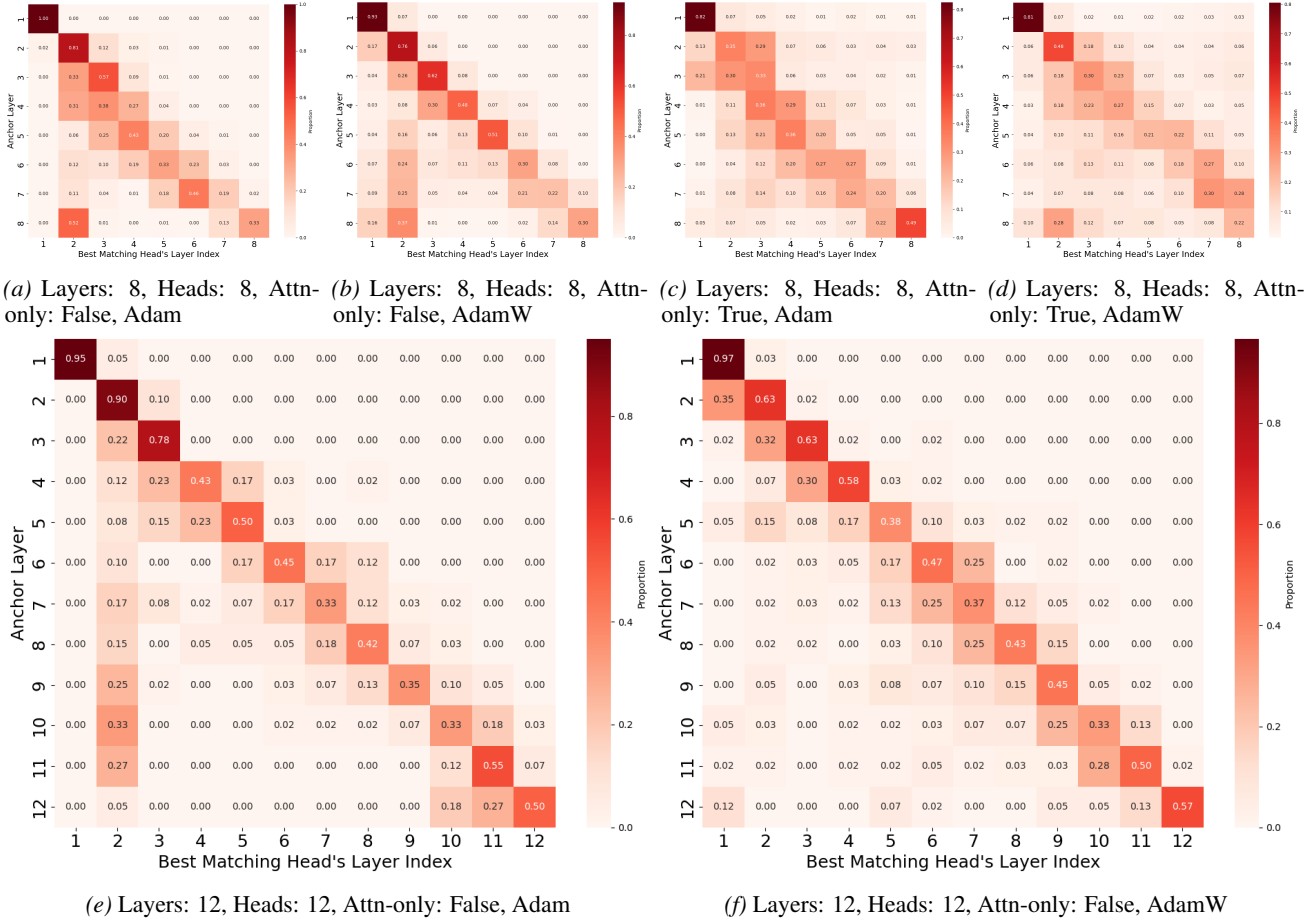

*(a)* Layers: 8, Heads: 8, Attn-only: False, Adam

*(b)* Layers: 8, Heads: 8, Attn-only: False, AdamW

*(c)* Layers: 8, Heads: 8, Attn-only: True, Adam

*(d)* Layers: 8, Heads: 8, Attn-only: True, AdamW

*(e)* Layers: 12, Heads: 12, Attn-only: False, Adam

*(f)* Layers: 12, Heads: 12, Attn-only: False, AdamW

*Figure 21.* Representative plots for alignment map & middle-layer dispersion § 4.2

## B.3. Within-layer uniqueness of attention heads

Representative plots for few of the architectures § 4.3:

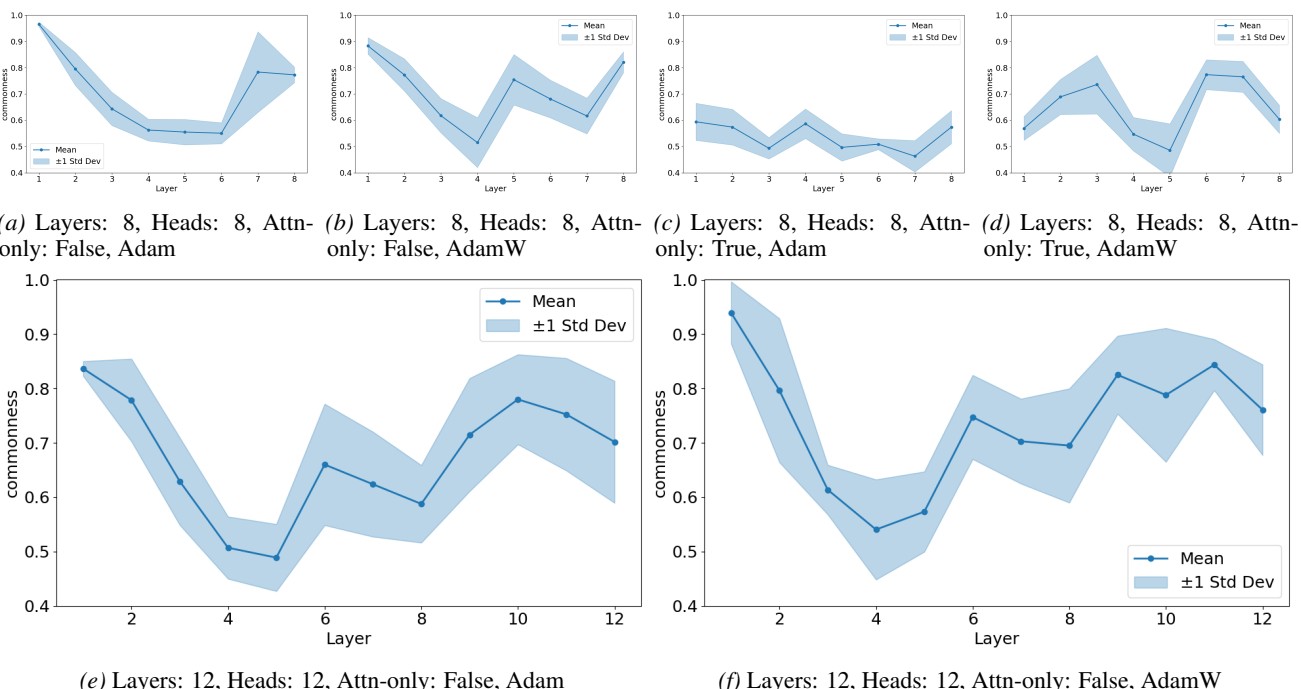

*(a)* Layers: 8, Heads: 8, Attn-only: False, Adam

*(b)* Layers: 8, Heads: 8, Attn-only: False, AdamW

*(c)* Layers: 8, Heads: 8, Attn-only: True, Adam

*(d)* Layers: 8, Heads: 8, Attn-only: True, AdamW

*(e)* Layers: 12, Heads: 12, Attn-only: False, Adam

*(f)* Layers: 12, Heads: 12, Attn-only: False, AdamW

*Figure 22.* Representative plots for cross-layer best-match stability § 4.3

## B.4. EFFECT OF PROMPT LENGTH ON STABILITY

Additional architecture-wise plots for § 4.4.2:

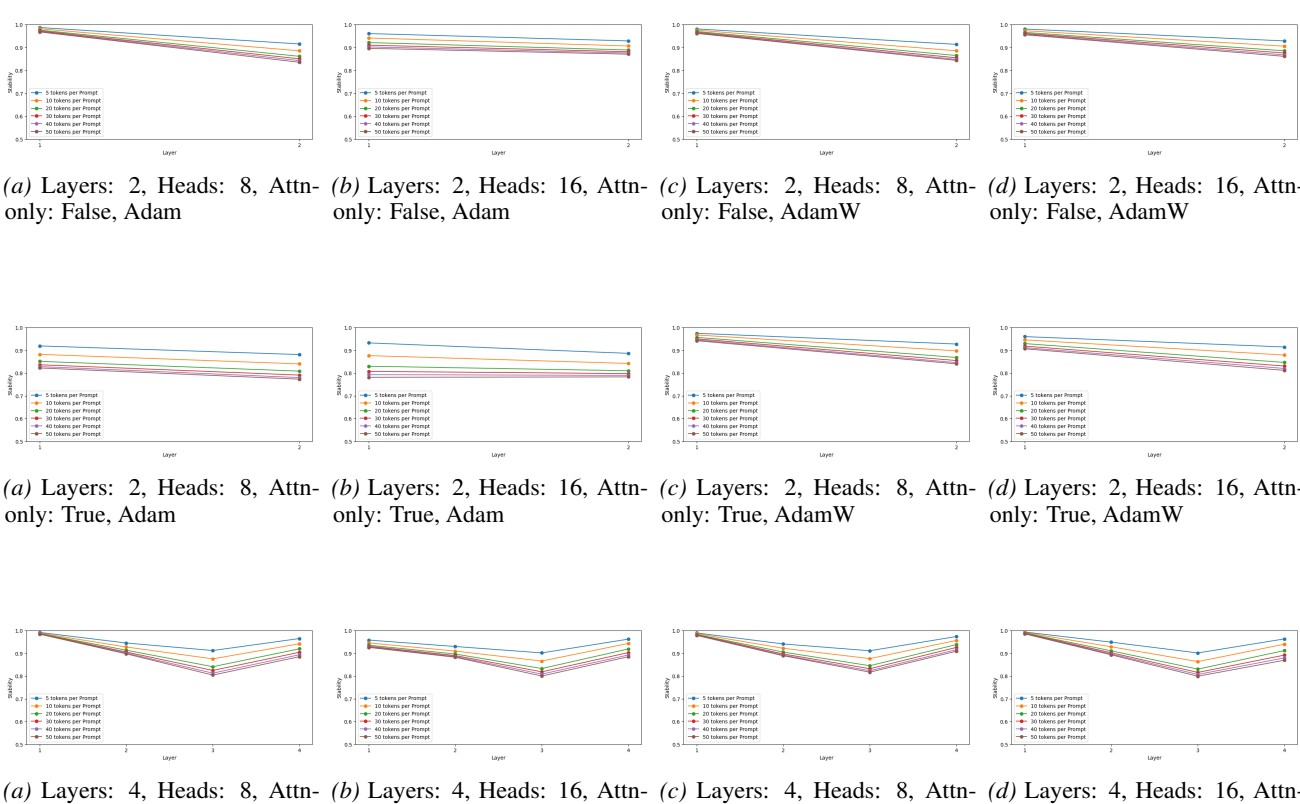

*(a)* Layers: 2, Heads: 8, Attn-only: False, Adam

*(b)* Layers: 2, Heads: 16, Attn-only: False, Adam

*(c)* Layers: 2, Heads: 8, Attn-only: False, AdamW

*(d)* Layers: 2, Heads: 16, Attn-only: False, AdamW

*(a)* Layers: 2, Heads: 8, Attn-only: True, Adam

*(b)* Layers: 2, Heads: 16, Attn-only: True, Adam

*(c)* Layers: 2, Heads: 8, Attn-only: True, AdamW

*(d)* Layers: 2, Heads: 16, Attn-only: True, AdamW

*(a)* Layers: 4, Heads: 8, Attn-only: False, Adam

*(b)* Layers: 4, Heads: 16, Attn-only: False, Adam

*(c)* Layers: 4, Heads: 8, Attn-only: False, AdamW

*(d)* Layers: 4, Heads: 16, Attn-only: False, AdamW

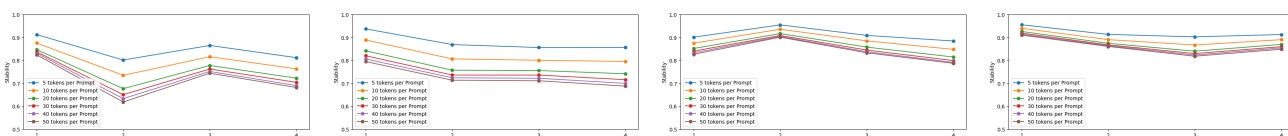

*(a)* Layers: 4, Heads: 8, Attn-only: True, Adam

*(b)* Layers: 4, Heads: 16, Attn-only: True, Adam

*(c)* Layers: 4, Heads: 8, Attn-only: True, AdamW

*(d)* Layers: 4, Heads: 16, Attn-only: True, AdamW

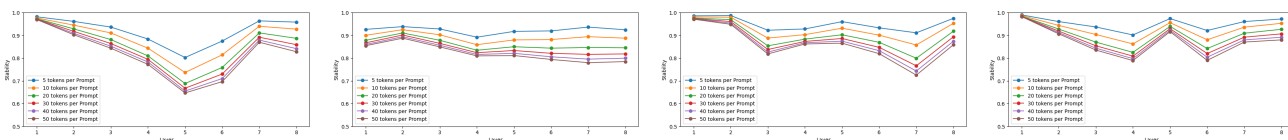

*(a)* Layers: 8, Heads: 8, Attn-only: False, Adam

*(b)* Layers: 8, Heads: 16, Attn-only: False, Adam

*(c)* Layers: 8, Heads: 8, Attn-only: False, AdamW

*(d)* Layers: 8, Heads: 16, Attn-only: False, AdamW

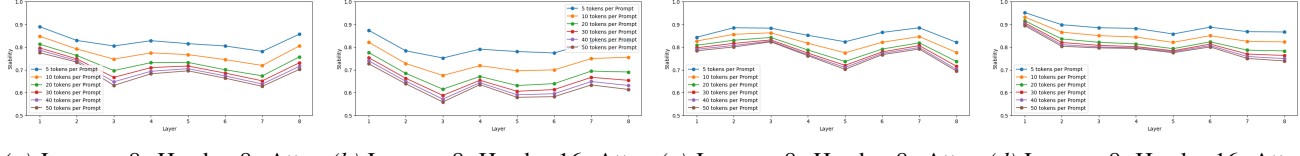

*(a)* Layers: 8, Heads: 8, Attn-only: True, Adam

*(b)* Layers: 8, Heads: 16, Attn-only: True, Adam

*(c)* Layers: 8, Heads: 8, Attn-only: True, AdamW

*(d)* Layers: 8, Heads: 16, Attn-only: True, AdamW

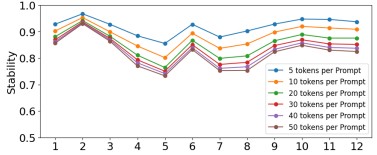

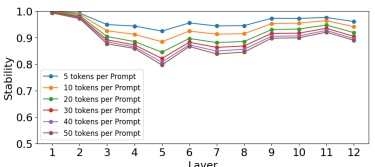

*(a)* Layers: 12, Heads: 12, Attn-only: False, Adam

*(b)* Layers: 12, Heads: 12, Attn-only: False, AdamW

## B.5. Correlation between Query-weight norm and layer-wise stability

| Layer | Heads | Attention-only | Optimizer | Correlation |
|---|---|---|---|---|
| 2 | 8 | False | False | -1.0 |
| 2 | 8 | False | True | -1.0 |
| 2 | 8 | True | False | -1.0 |
| 2 | 8 | True | True | -1.0 |
| 2 | 16 | False | False | -1.0 |
| 2 | 16 | False | True | -1.0 |
| 2 | 16 | True | False | 1.0 |
| 2 | 16 | True | True | -1.0 |
| 4 | 8 | False | False | -0.9406589554922179 |
| 4 | 8 | False | True | -0.9015930143944082 |
| 4 | 8 | True | False | -0.6827415649248102 |
| 4 | 8 | True | True | -0.2212791321322467 |
| 4 | 16 | False | False | -0.8397480100370013 |
| 4 | 16 | False | True | -0.9434888997439926 |
| 4 | 16 | True | False | -0.6494382579338498 |
| 4 | 16 | True | True | -0.9554985039029008 |
| 8 | 8 | False | False | -0.8447143648189646 |
| 8 | 8 | False | True | -0.6931182527732535 |
| 8 | 8 | True | False | -0.7659110184144368 |
| 8 | 8 | True | True | -0.031545835341075135 |
| 8 | 16 | False | False | -0.21617605236079285 |
| 8 | 16 | False | True | -0.6749555081511857 |
| 8 | 16 | True | False | -0.6941387312838879 |
| 8 | 16 | True | True | -0.6374924846114093 |
| 12 | 12 | False | False | -0.5078991103274401 |
| 12 | 12 | False | True | -0.8228023018292414 |

*Figure 30.* Layer-wise stability is inversely correlated with average ($\ell_2$) norm of attention-head query weights, suggesting a link between norm growth and head instability.

## B.6. Stability comparison: Adam vs AdamW

Additional architecture-wise plots showing improvement in stability when AdamW optimizer is used while pre-training (§ 4.5):

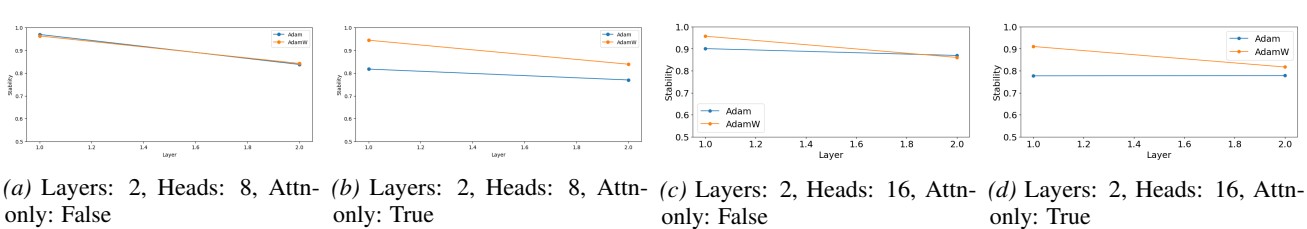

*(a)* Layers: 2, Heads: 8, Attn-only: False  *(b)* Layers: 2, Heads: 8, Attn-only: True  *(c)* Layers: 2, Heads: 16, Attn-only: False  *(d)* Layers: 2, Heads: 16, Attn-only: True

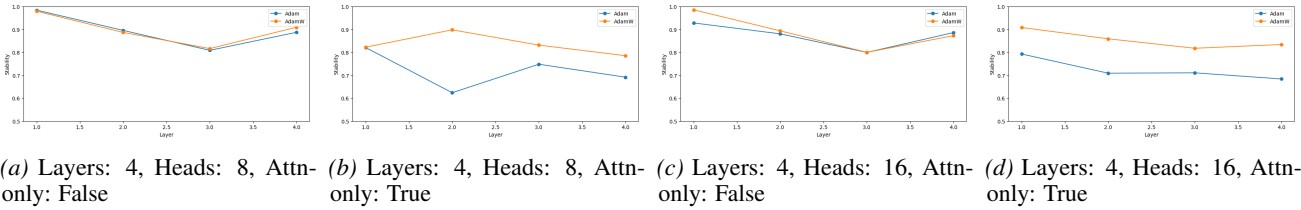

*(a)* Layers: 4, Heads: 8, Attn-only: False

*(b)* Layers: 4, Heads: 8, Attn-only: True

*(c)* Layers: 4, Heads: 16, Attn-only: False

*(d)* Layers: 4, Heads: 16, Attn-only: True

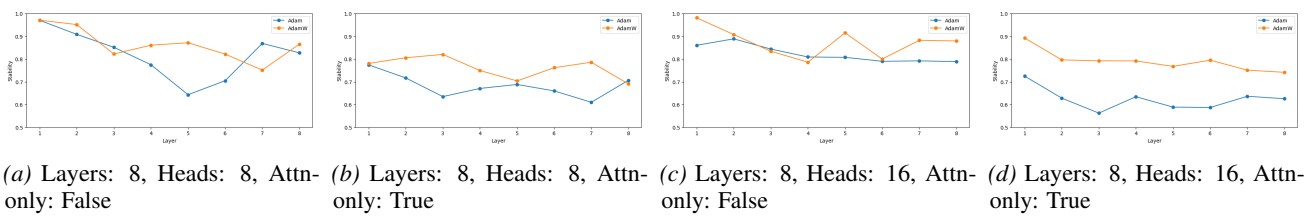

*(a)* Layers: 8, Heads: 8, Attn-only: False

*(b)* Layers: 8, Heads: 8, Attn-only: True

*(c)* Layers: 8, Heads: 16, Attn-only: False

*(d)* Layers: 8, Heads: 16, Attn-only: True

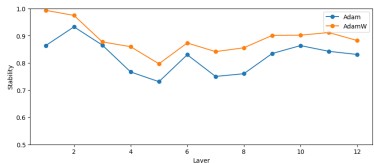

*(a)* Layers: 12, Heads: 12, Attn-only: False

### B.6.1. MECHANISTIC CHECK

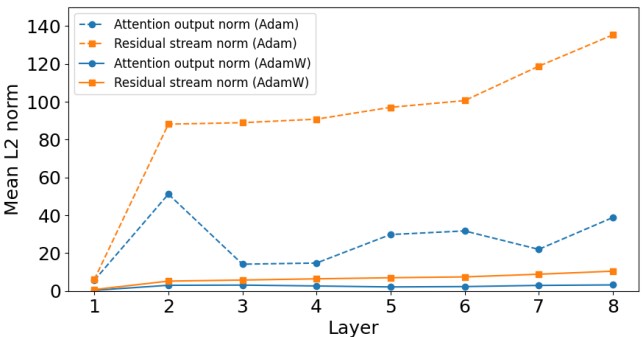

*Figure 35.* AdamW substantially controls norm growth. See § 4.5

## B.7. Performance Parity: Adam vs AdamW

Additional architecture-wise plots showing **Performance Parity** between Adam and AdamW variants as mentioned in § 4.5. The validation perplexity was calculated using dataset "wikitext-2-raw-v1" (Merity et al., 2016):

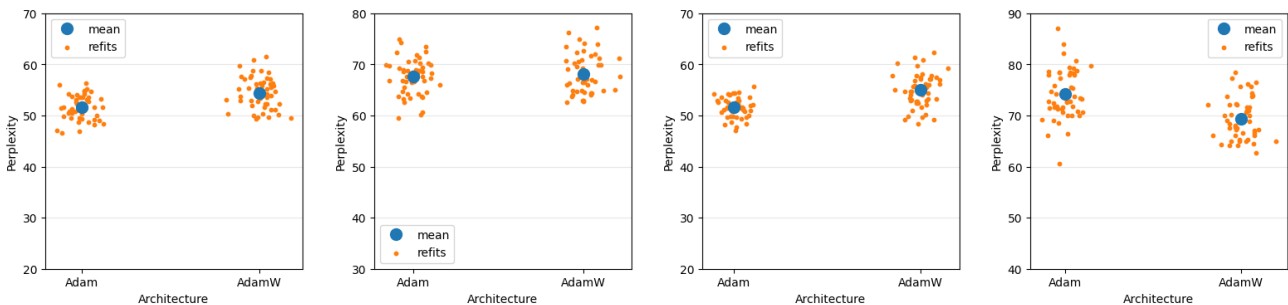

*(a)* Layers: 2, Heads: 8, Attn-only: False

*(b)* Layers: 2, Heads: 8, Attn-only: True

*(c)* Layers: 2, Heads: 16, Attn-only: False

*(d)* Layers: 2, Heads: 16, Attn-only: True

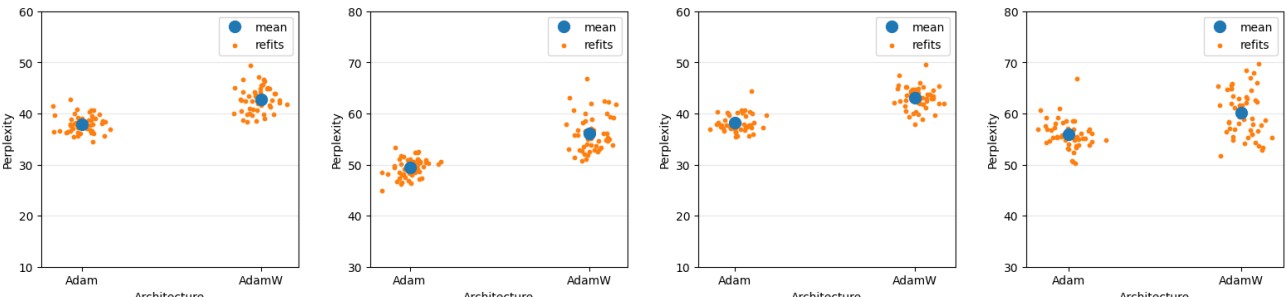

*(a)* Layers: 4, Heads: 8, Attn-only: False

*(b)* Layers: 4, Heads: 8, Attn-only: True

*(c)* Layers: 4, Heads: 16, Attn-only: False

*(d)* Layers: 4, Heads: 16, Attn-only: True

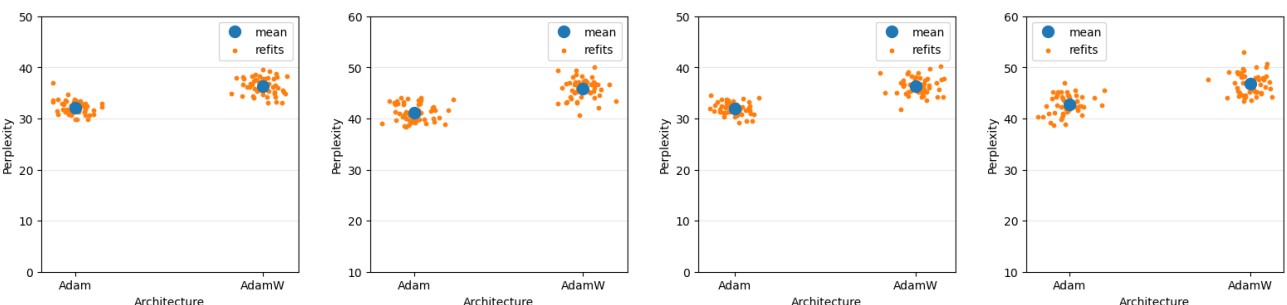

*(a)* Layers: 8, Heads: 8, Attn-only: False

*(b)* Layers: 8, Heads: 8, Attn-only: True

*(c)* Layers: 8, Heads: 16, Attn-only: False

*(d)* Layers: 8, Heads: 16, Attn-only: True

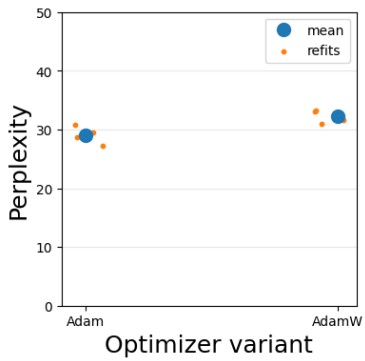

*(a)* Layers: 12, Heads: 12, Attn-only: False

## B.8. Stability of residual stream

Additional architecture-wise plots for § 4.7:

*(a)* Layers: 2, Heads: 8, Attn-only: False
*(b)* Layers: 2, Heads: 8, Attn-only: True
*(c)* Layers: 2, Heads: 16, Attn-only: False
*(d)* Layers: 2, Heads: 16, Attn-only: True

*(a)* Layers: 4, Heads: 8, Attn-only: False
*(b)* Layers: 4, Heads: 8, Attn-only: True
*(c)* Layers: 4, Heads: 16, Attn-only: False
*(d)* Layers: 4, Heads: 16, Attn-only: True

*(a)* Layers: 8, Heads: 8, Attn-only: False
*(b)* Layers: 8, Heads: 8, Attn-only: True
*(c)* Layers: 8, Heads: 16, Attn-only: False
*(d)* Layers: 8, Heads: 16, Attn-only: True

*(a)* Layers: 12, Heads: 12, Attn-only: False

## B.9. Correlation between the stability and "post-ablation change in perplexity"

Additional architecture-wise plots for § 4.8:

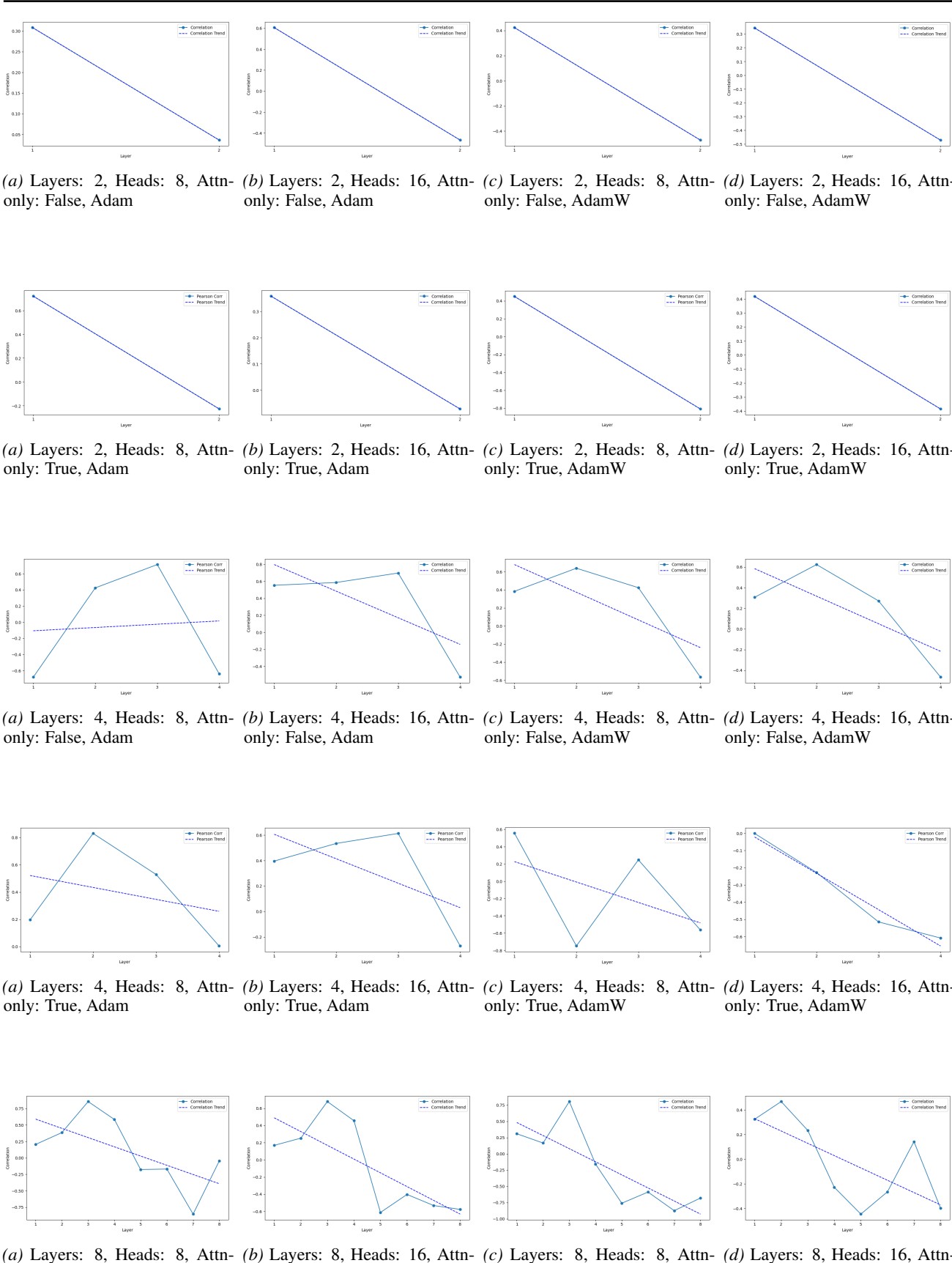

*(a)* Layers: 2, Heads: 8, Attn-only: False, Adam

*(b)* Layers: 2, Heads: 16, Attn-only: False, Adam

*(c)* Layers: 2, Heads: 8, Attn-only: False, AdamW

*(d)* Layers: 2, Heads: 16, Attn-only: False, AdamW

*(a)* Layers: 2, Heads: 8, Attn-only: True, Adam

*(b)* Layers: 2, Heads: 16, Attn-only: True, Adam

*(c)* Layers: 2, Heads: 8, Attn-only: True, AdamW

*(d)* Layers: 2, Heads: 16, Attn-only: True, AdamW

*(a)* Layers: 4, Heads: 8, Attn-only: False, Adam

*(b)* Layers: 4, Heads: 16, Attn-only: False, Adam

*(c)* Layers: 4, Heads: 8, Attn-only: False, AdamW

*(d)* Layers: 4, Heads: 16, Attn-only: False, AdamW

*(a)* Layers: 4, Heads: 8, Attn-only: True, Adam

*(b)* Layers: 4, Heads: 16, Attn-only: True, Adam

*(c)* Layers: 4, Heads: 8, Attn-only: True, AdamW

*(d)* Layers: 4, Heads: 16, Attn-only: True, AdamW

*(a)* Layers: 8, Heads: 8, Attn-only: False, Adam

*(b)* Layers: 8, Heads: 16, Attn-only: False, Adam

*(c)* Layers: 8, Heads: 8, Attn-only: False, AdamW

*(d)* Layers: 8, Heads: 16, Attn-only: False, AdamW

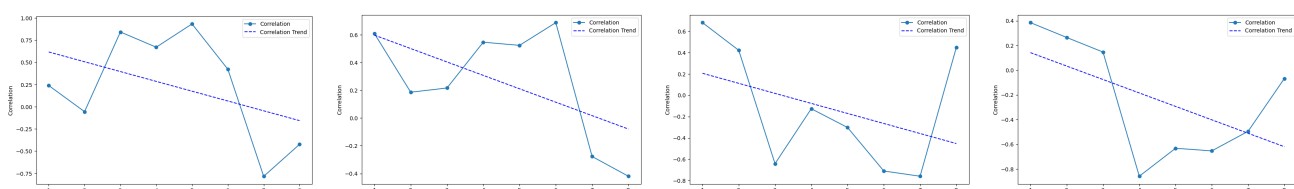

*(a)* Layers: 8, Heads: 8, Attn-only: True, Adam

*(b)* Layers: 8, Heads: 16, Attn-only: True, Adam

*(c)* Layers: 8, Heads: 8, Attn-only: True, AdamW

*(d)* Layers: 8, Heads: 16, Attn-only: True, AdamW

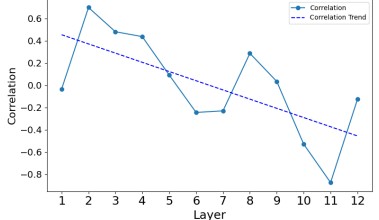

*(a)* Layers: 12, Heads: 12, Attn-only: False, Adam

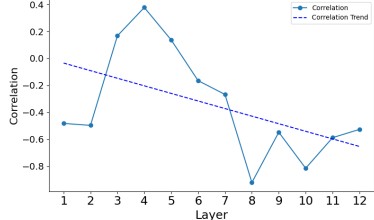

*(b)* Layers: 12, Heads: 12, Attn-only: False, AdamW

*Listing 1.* TransformerLens HookedTransformerConfig template for MLP architectures

```
1  {
2  'act_fn': 'gelu',
3   'attention_dir': 'causal',
4   'attn_only': False,
5   'attn_scale': np.float64(8.0),
6   'attn_scores_soft_cap': -1.0,
7   'attn_types': None,
8   'checkpoint_index': None,
9   'checkpoint_label_type': None,
10   'checkpoint_value': None,
11   'd_head': 64,
12   'd_mlp': 2048,
13   'd_model': 512,
14   'd_vocab': 48262,
15   'd_vocab_out': 48262,
16   'decoder_start_token_id': None,
17   'default_prepend_bos': True,
18   'device': device(type='cuda'),
19   'dtype': torch.float32,
20   'eps': 1e-05,
21   'experts_per_token': None,
22   'final_rms': False,
23   'from_checkpoint': False,
24   'gated_mlp': False,
25   'init_mode': 'gpt2',
26   'init_weights': True,
27   'initializer_range': np.float64(0.035355339059327376),
28   'load_in_4bit': False,
29   'model_name': 'GELU_8L512W_C4_Code_8H',
30   'n_ctx': 1024,
31   'n_devices': 1,
32   'n_heads': [NUM\_HEADS],
33   'n_key_value_heads': None,
34   'n_layers': [NUM\_LAYERS],
35   'n_params': 25165824,
36   'normalization_type': 'LN',
37   'num_experts': None,
38   'original_architecture': None,
39   'output_logits_soft_cap': -1.0,
40   'parallel_attn_mlp': False,
41   'positional_embedding_type': 'standard',
42   'post_embedding_ln': False,
43   'relative_attention_max_distance': None,
44   'relative_attention_num_buckets': None,
45   'rotary_adjacent_pairs': False,
46   'rotary_base': 10000,
47   'rotary_dim': None,
48   'scale_attn_by_inverse_layer_idx': False,
49   'seed': 2,
50   'tie_word_embeddings': False,
51   'tokenizer_name': 'NeelNanda/gpt-neox-tokenizer-digits',
52   'tokenizer_prepends_bos': False,
53   'trust_remote_code': False,
54   'use_attn_in': False,
55   'use_attn_result': False,
56   'use_attn_scale': True,
57   'use_hook_mlp_in': False,
58   'use_hook_tokens': False,
59   'use_local_attn': False,
60   'use_normalization_before_and_after': False,
61   'use_split_qkv_input': False,
62   'window_size': None
63  }
```

*Listing 2.* TransformerLens HookedTransformerConfig template for Attention-only architectures

```
 1  {
 2  'act_fn': 'gelu',
 3   'attention_dir': 'causal',
 4   'attn_only': True,
 5   'attn_scale': np.float64(8.0),
 6   'attn_scores_soft_cap': -1.0,
 7   'attn_types': None,
 8   'checkpoint_index': None,
 9   'checkpoint_label_type': None,
10   'checkpoint_value': None,
11   'd_head': 64,
12   'd_mlp': 2048,
13   'd_model': 512,
14   'd_vocab': 48262,
15   'd_vocab_out': 48262,
16   'decoder_start_token_id': None,
17   'default_prepend_bos': True,
18   'device': device(type='cuda'),
19   'dtype': torch.float32,
20   'eps': 1e-05,
21   'experts_per_token': None,
22   'final_rms': False,
23   'from_checkpoint': False,
24   'gated_mlp': False,
25   'init_mode': 'gpt2',
26   'init_weights': True,
27   'initializer_range': np.float64(0.035355339059327376),
28   'load_in_4bit': False,
29   'model_name': 'GELU_8L512W_C4_Code_8H_Attn_only',
30   'n_ctx': 1024,
31   'n_devices': 1,
32   'n_heads': [NUM\_HEADS],
33   'n_key_value_heads': None,
34   'n_layers': [NUM\_LAYERS],
35   'n_params': 8388608,
36   'normalization_type': 'LN',
37   'num_experts': None,
38   'original_architecture': None,
39   'output_logits_soft_cap': -1.0,
40   'parallel_attn_mlp': False,
41   'positional_embedding_type': 'standard',
42   'post_embedding_ln': False,
43   'relative_attention_max_distance': None,
44   'relative_attention_num_buckets': None,
45   'rotary_adjacent_pairs': False,
46   'rotary_base': 10000,
47   'rotary_dim': None,
48   'scale_attn_by_inverse_layer_idx': False,
49   'seed': 4,
50   'tie_word_embeddings': False,
51   'tokenizer_name': 'NeelNanda/gpt-neox-tokenizer-digits',
52   'tokenizer_prepends_bos': False,
53   'trust_remote_code': False,
54   'use_attn_in': False,
55   'use_attn_result': False,
56   'use_attn_scale': True,
57   'use_hook_mlp_in': False,
58   'use_hook_tokens': False,
59   'use_local_attn': False,
60   'use_normalization_before_and_after': False,
61   'use_split_qkv_input': False,
62   'window_size': None
63  }
```

*Listing 3.* TransformerLens HookedTransformerConfig template for GPT2-small architectures

```
1  {
2  'act_fn': 'gelu',
3   'attention_dir': 'causal',
4   'attn_only': False,
5   'attn_scale': np.float64(8.0),
6   'attn_scores_soft_cap': -1.0,
7   'attn_types': None,
8   'checkpoint_index': None,
9   'checkpoint_label_type': None,
10  'checkpoint_value': None,
11  'd_head': 64,
12  'd_mlp': 3072,
13  'd_model': 768,
14  'd_vocab': 50257,
15  'd_vocab_out': 50257,
16  'decoder_start_token_id': None,
17  'default_prepend_bos': True,
18  'device': device(type='cuda'),
19  'dtype': torch.float32,
20  'eps': 1e-05,
21  'experts_per_token': None,
22  'final_rms': False,
23  'from_checkpoint': False,
24  'gated_mlp': False,
25  'init_mode': 'gpt2',
26  'init_weights': True,
27  'initializer_range': np.float64(0.02886751345948129),
28  'load_in_4bit': False,
29  'model_name': 'gpt2',
30  'n_ctx': 1024,
31  'n_devices': 1,
32  'n_heads': 12,
33  'n_key_value_heads': None,
34  'n_layers': 12,
35  'n_params': 84934656,
36  'normalization_type': 'LN',
37  'num_experts': None,
38  'original_architecture': None,
39  'output_logits_soft_cap': -1.0,
40  'parallel_attn_mlp': False,
41  'positional_embedding_type': 'standard',
42  'post_embedding_ln': False,
43  'relative_attention_max_distance': None,
44  'relative_attention_num_buckets': None,
45  'rotary_adjacent_pairs': False,
46  'rotary_base': 10000,
47  'rotary_dim': None,
48  'scale_attn_by_inverse_layer_idx': False,
49  'seed': 4,
50  'tie_word_embeddings': False,
51  'tokenizer_name': 'gpt2',
52  'tokenizer_prepends_bos': False,
53  'trust_remote_code': False,
54  'use_attn_in': False,
55  'use_attn_result': False,
56  'use_attn_scale': True,
57  'use_hook_mlp_in': False,
58  'use_hook_tokens': False,
59  'use_local_attn': False,
60  'use_normalization_before_and_after': False,
61  'use_split_qkv_input': False,
62  'window_size': None
63  }
```

*Listing 4.* TransformerLens HookedTransformerTrainConfig template for architectures with number of layers in (2,4,8) and trained using Adam optimizer

```
{
  "batch_size": 5,
  "lr": 0.0001,
  "optimizer_name": "Adam",
  "weight_decay": None,
  "num_epochs": 1,
  "save_every": 66285
}
```

*Listing 5.* TransformerLens HookedTransformerTrainConfig template for architectures with number of layers in (2,4,8) and trained using AdamW optimizer

```
{
  "batch_size": 5,
  "lr": 1e-4,
  "optimizer_name": "AdamW",
  "weight_decay": 0.1,
  "num_epochs": 1,
  "save_every": 66285
}
```

*Listing 6.* TransformerLens HookedTransformerTrainConfig template for gpt2-small architectures and trained using Adam optimizer

```
{
  "batch_size": 5,
  "lr": 5e-05,
  "optimizer_name": "Adam",
  "weight_decay": None,
  "num_epochs": 1,
  "save_every": 66285
}
```

*Listing 7.* TransformerLens HookedTransformerTrainConfig template for gpt2-small architectures and trained using AdamW optimizer

```
{
  "batch_size": 5,
  "lr": 1e-4,
  "optimizer_name": "AdamW",
  "weight_decay": 0.1,
  "num_epochs": 1,
  "save_every": 66285
}
```

