# OpenReview forum: "Quantifying LLM Attention-Head Stability: Implications for Circuit Universality"
_ICML.cc/2026/Conference — ICML 2026 regular_

### Official Review · Reviewer_9aBk · 2026-03-01

**Soundness:** 3
**Presentation:** 3
**Significance:** 4
**Originality:** 3
**Overall Recommendation:** 5
**Confidence:** 3

**Summary:**

This paper examines whether attention heads in independently trained refits of the same architecture are stable across random seeds. Using a permutation-invariant metric based on attention score matching, it identifies a consistent mid-layer stability dip, where heads are least stable yet most unique across refits. It further shows that deeper models amplify this divergence, that unstable heads in deeper layers become more functionally important, and that AdamW improves stability with minimal performance cost, underscoring the need for cross-refit validation of circuit claims.

**Compliance With Llm Reviewing Policy:**

Affirmed.

**Key Questions For Authors:**

See weakness

**Limitations:**

Yes

**Strengths And Weaknesses:**

### **Strengths**

**S1:** The paper raises an important question about cross-seed stability and its implications for claims of universality in mechanistic interpretability; the framing is timely and well motivated.

**S2:** The methodology is clean and principled. Measuring head similarity via attention score matrices and using best-match aggregation provides a clear, operational definition of seed stability.

**S3:** The study covers diverse architectural configurations, including both MLP and attention-only variants, making the empirical patterns more convincing than a single-architecture analysis.

------

### **Weaknesses**

**W1:** While the 2–8-layer models use 50 refits, the 12-layer GPT-2-small setting includes only 5. Since several key conclusions concern depth dependence, the evidence in the deepest regime is less statistically robust; a more rigorous evaluation with additional refits would strengthen the claims if computational budget permits.

**W2:** The stability metric relies on attention score matrices. Although interpretable, it may overlook cases where heads differ in attention patterns yet contribute similarly via value projections or downstream residual interactions, leaving part of the functional similarity unexamined.

---

> ### Author Rebuttal · Authors · 2026-03-31
>
> We thank the reviewer for the thoughtful comments, positive evaluation and the recommendation for acceptance. We found the feedback helpful and have used it to clarify the identified weaknesses (**W**).
>
> \
> **W1: “[...] GPT-2-small setting includes only 5 [...]”  (scaling: number of refits, model depth, architectural diversity)**
>
> This is a joint response for concerns related to “scaling up”:
>
> We agree that our models are smaller than current frontier LLMs. We also agree that increasing the number of GPT-2-small refits would strengthen the study. However, we had to balance scaling along three competing dimensions: the number of refits, model depth, and architectural diversity. Given a fixed computational budget, expanding one of these dimensions necessarily limited the others.
>
> For concern regarding model size, toy transformers and GPT-2-scale models remain standard and widely accepted in mechanistic interpretability research because they allow controlled experimentation and repeated refits at manageable cost [1][2][3]. This is the setting used in several foundational papers in the area.
>
> For concern regarding refits, our decision to train 10 GPT-2 instances (for Adam and AdamW) was guided by prior related works [4] on stability across training runs.
>
> For example, recent related work on stability across training runs has also examined models in a comparable parameter regime. For example, [4] studies 5 sets of models ranging from 14M to 410M parameters, with 9 independent training seeds per model. In comparison, we train 1,210 independent refits from scratch across 26 architectural configurations, a feat that would be unattainable for 7B+ parameter models.
>
> While many mechanistic interpretability studies do investigate models of this size, they almost exclusively study either fully pre-trained or fine-tuned models. Our key contribution is the analysis of attention-based circuit universality across pre-training refits; we would not be able to study this phenomenon to any degree of statistical certainty if we pre-trained 7B+ parameter LLMs from scratch.
>
> [1] A Mathematical Framework for Transformer Circuits (Elhage et al., 2021)\
> [2] In-context Learning and Induction Heads (Olsson et al., 2022)\
> [3] Emergent World Representations: Exploring a Sequence Model Trained on a Synthetic Task (Li et al., 2022 / ICLR 2023)\
> [4] PolyPythias: Stability and Outliers across Fifty Language Model Pre-Training Runs (Wal et al., 2025 / ICLR 2025)
>
> \
> **W2: “[...] The stability metric [...]”  (use of attention score)**
>
> We agree with the broader point that the attention score should not automatically be interpreted as a faithful explanation of feature importance, or the attention head as a whole, due to the ability of heads with different attention patterns to contribute similarly. This is a well-known debate in the interpretability literature, with arguments both for [1] and against [2][3] that view. In our work, however, attention scores are not used as an explanation signal for token importance or a head’s functionality. Rather, we use the post-softmax attention matrix as a behavioural signature of a head’s routing pattern: namely, which positions a head mixes information from, and how stable that routing remains across random-seed refits. We maintain that this signature is a key part of a head’s identity, regardless of how similarly or differently it manifests in downstream contributions.
>
> This framing is aligned with a large body of mechanistic interpretability work, where attention patterns are used to identify functional heads and circuits such as induction heads [4], previous-token heads, and related routing behaviours [5]. Similarly, works in critical areas have treated attention scores as a core analytical signal [6]. Our claim is therefore not that “attention score is explanation,” but that “attention score is a useful and measurable signature of routing behaviour”. And depending upon context, its stability across refits is itself informative.
>
> [1] Attention is not Explanation — Wallace et al. (2019)\
> [2] Attention Interpretability Across NLP Tasks — Vashishth et al. (2019)\
> [3] What Does BERT Look at? An Analysis of BERT’s Attention — Clark et al. (2019)\
> [4] In-context Learning and Induction Heads — Olsson et al. (2022)\
> [5] Kamath, Ameisen, et al., "Tracing Attention Computation", Transformer Circuits, 2025.\
> [6] BERTology Meets Biology: Interpreting Attention in Protein Language Models - J Vig et al.  (2020)

---

> > ### Author Rebuttal · Reviewer_9aBk · 2026-03-31
> >
> > My concern about W2 has been addressed. For W1, the authors explain the motivation behind their choice in the text, but they do not provide additional experiments to support the claim, so I consider it only partially resolved.

---

> > > ### Author Response · Authors · 2026-04-04
> > >
> > > We thank the reviewer for the thoughtful additional comments and for acknowledging that our earlier clarification addressed many of their concerns.
> > >
> > > **W1**
> > >
> > > In addition to the clarifications provided in our first response, we would like to respectfully note that we do not plan to add new results at this stage by training additional refits on the largest model sizes evaluated in the paper. Such experiments would require additional time beyond what is available during the current rebuttal stage.

---

### Official Review · Reviewer_GWfe · 2026-03-06

**Soundness:** 4
**Presentation:** 3
**Significance:** 4
**Originality:** 3
**Overall Recommendation:** 5
**Confidence:** 5

**Summary:**

This is one of the most interesting papers I am reviewing in this cycle. The mechanistic study asks whether circuits identified in mechanistic interpretability are robust across independent refits. This is a timely work that calls us to more carefully examine our assumptions about attention heads in decoder-only language models. The authors train the same architecture multiple times with different random seeds and directly measure how stable individual attention heads are across refits. While the idea is simple, the conclusions it yields are profound. They define a stability metric based on cosine similarity between post-softmax attention score matrices and ask how similar the most similar head in another refit is to a given attention head. Across 26 architectural variants, they find a pronounced mid-layer stability dip, showing that even models trained with identical data and hyperparameters can diverge substantially in their internal structure. Building on this core finding, the paper shows that instability is systematically related to depth, representational uniqueness, and optimizer choice, revealing that the internal circuits we often analyze may be far less universal than prior literature on attention heads may suggest.

**Compliance With Llm Reviewing Policy:**

Affirmed.

**Final Justification:**

I maintain my accept score. Cross-refit attention-head stability is a timely and important question for mechanistic interpretability, and this paper addresses it carefully. The empirical study is well executed, and the finding that internal attention-head structure can diverge across refits despite identical setup worth sharing with the community.

**Key Questions For Authors:**

I understood the content sufficiently.

**Limitations:**

Yes.

**Strengths And Weaknesses:**

**Soundness:** The paper is technically sound and carefully executed. The stability metric is clearly defined and consistently applied across a large number of refits and architectural variants. The authors control key training factors and conduct extensive ablations, including analyses of depth effects, prompt length sensitivity, optimizer choice, residual-stream stability, and head-level functional importance. They also present results transparently, even when those results complicate or qualify the central narrative. Overall, the empirical foundation of the paper’s claims is strong.

**Presentation:** The paper is clearly written and logically organized. The motivation, metric definition, and experimental pipeline are presented in a coherent progression, and the appendices are thorough and provide sufficient detail for reproduction. There are a few minor presentational issues. For example, Figure 11 refers to a “ratio,” whereas the caption uses the term “depth,” and in Figure 2 the color legend in the top panel could be truncated at 0.5 to better match the bottom panel. These are minor cosmetic issues and are far outweighed by the overall clarity and comprehensiveness of the presentation. As a point of personal curiosity, it would also be interesting to see a short qualitative example contrasting specific textual attention patterns across refits in highly unstable layers, as this might offer additional intuition alongside the quantitative results, though this could naturally be left to future work.

**Originality:** The originality of this work lies in applying well-established tools to a question that has been largely overlooked in the literature. By examining attention heads through the lens of cross-refit stability, the paper introduces a missing perspective in mechanistic interpretability and derives conclusions that are both surprising and consequential.

**Significance:**  The significance of this work lies in showing that internal structure can diverge meaningfully despite training on the same architecture and the same data. This is profound because much of the prior literature on attention heads implicitly treats them as relatively stable, functionally persistent units of computation. This finding should influence how interpretability results are validated and how claims about circuits are framed going forward.

---

> ### Author Rebuttal · Authors · 2026-03-31
>
> We thank the reviewer for the positive evaluation and the recommendation for acceptance. We are pleased that the reviewer found the paper clear and did not raise major questions or concerns about the technical content or empirical validation. We appreciate the reviewer’s support and will further improve the presentation in the final version for clarity and completeness.

---

> > ### Author Rebuttal · Reviewer_GWfe · 2026-03-31
> >
> > I appreciate the response and maintain my accept score.

---

### Official Review · Reviewer_xxMp · 2026-03-08

**Soundness:** 3
**Presentation:** 3
**Significance:** 3
**Originality:** 3
**Overall Recommendation:** 4
**Confidence:** 4

**Summary:**

The authors test if the attention probability matrices of models trained with different seeds converge to being the same. They run experiments on 26 models of varying sizes. They conclude that the stability of heads depends on different factors like depth or optimizer.

**Compliance With Llm Reviewing Policy:**

Affirmed.

**Final Justification:**

The authors addressed my primary concern, regarding normalizing the metric by the sequence length, and therefore, I have increased my score during the rebuttal to weak accept.

**Key Questions For Authors:**

- In Section 4.4.2, you conclude that stability decreases sharply as prompt length increases. However, because causal masking forces the first token to attend entirely to itself, the unnormalized similarity metric (Equation 1) has a mathematical lower bound of 1/T for a prompt of length T. As T increases from 5 to 50, this floor drops from 0.20 to 0.02. Could the drop in stability simply be a mathematical artifact of the shifting lower bound rather than a mechanistic property of the heads? If you could demonstrate that the drop in stability remains after applying a normalization to the metric, I would be much more confident in this claim. If not, Section 4.4.2 needs revision.
- The primary evaluation set consists of 100 prompts ranging from 15 to 80 tokens, generated by ChatGPT. Given the mathematical lower bound issue of the stability metric mentioned above, averaging unnormalized stability scores with different lower bounds makes the majority of the results difficult to interpret. Could you provide more justification for the current choice of metric, or rerun the relevant experiments with either a fixed prompt length or normalized stability scores?
- The proposed stability metric relies entirely on post-softmax attention probabilities. While this captures where information is routed, it ignores the value and output matrices, which determine what information is actually extracted and written to the residual stream. Two heads might have identical attention maps but their function could be very different, meaning the metric may produce false positives for circuit universality. How do you mitigate the risk of false positives with your metric? Could you provide empirical evidence or theoretical justification showing that matching attention routing highly correlates with matching value/output feature extraction in these models? This would alleviate my concern about the metric acting as an incomplete proxy for circuit universality and significantly strengthen the paper’s main claims.
- Why use cosine similarity between vectorized stochastic matrices as opposed to a metric that would rely on the fact that probabilities are being compared?
- Is the matching process equivalent to optimal transport? If so, why dont you use an off-the-shelf optimal transport solver?
- Can two heads in model A be matched to a single head in model B? Is this desirable or expected?

**Limitations:**

No. Please add the mathematical limitations of the proposed stability metric to the Limitations section.

**Strengths And Weaknesses:**

Strengths
- Propose a practical work-around to the problem of permutation symmetries present in neural networks. By comparing post-softmax attention probabilities, the authors established a shared, immutable basis for cross-seed comparison.
- Challenging the assumption that reported LLM circuits are universally robust is a good contribution and their results provide some clarity.
- Finding AdamW improves head stability, and potentially circuit universality, without sacrificing performance provides an actionable solution to improve mechanistic interpretability research across labs.
- The paper is well-written and structured. The main idea is well motivated and the narrative is easy to follow.


Weakness
- There is a significant mathematical issue with the proposed similarity metric (Equation 1). Because causal masking forces the first token to attend only to itself, the mathematical lower bound of the similarity score for a prompt of length T is 1/T. The upper bound is always 1. Because this metric is not normalized to account for the shifting floor, many of the results in Section 4 are a bit unclear. First, the primary evaluation set consists of 100 prompts ranging from 15-80 tokens. Averaging unnormalized similarity scores across prompts with different lower bounds makes the aggregate stability scores difficult to interpret. Second, in Section 4.4.2 the authors argue that greater prompt length leads to greater head instability. However, this result is heavily confounded by the metric’s bounds. As T increases from 5 to 50, the mathematical floor drops from 0.20 to 0.02. The observed drop in stability with increased prompt length may simply be an artifact of this. The authors should either ensure the prompt length is fixed or normalize the similarity scores properly.
- The chosen stability metric only considers the post-softmax attention probabilities. While this effectively measures where the model is routing information, it ignores the value and output matrices, which determine what information is actually being extracted and written to the residual stream. Two heads might have identical attention maps but their function could be very different, meaning the metric may produce false positives for circuit universality.
- The number of parameters in the models is small compared to standard LLM parameter counts (>1B param). The depth is also small compared to standard LLM models. This raises questions as to whether the conclusions would hold for larger models.
- “fig” not capitalized in 275. Capital letter issue in caption of Fig. 2.
- Subsection 4.4.1 is unclear and refers to a figure in the appendix.
- Weren’t there previous works that have already identified the stability of the residual stream across depth, as observed in section 4.7?
- The paper reads like a series of observations but lacks a narrative that would tie them into a cohesive story.

---

> ### Author Rebuttal · Authors · 2026-03-31
>
> We thank the reviewer for the thoughtful and constructive comments. We found the feedback very helpful and used it to clarify the noted weakness (**W**) and address the questions raised (**Q**).
>
> \
> **W1 / Q1 (“In Section 4.4.2 [...]”) and Q2 (“[...] 100 prompts ranging from 15 to 80 tokens [...]”)**
>
> We understand the reviewer’s concern about possible bias when comparing prompts of different lengths. However, that concern does not apply to our methods as used here. We do not compare attention representations across prompts of different lengths. Instead, for a given i’th prompt, we compare its attention-score representation across multiple instances (seeds).
>
> Causal masking and length dependence are intrinsic to autoregressive transformers. If longer prompts reduce cross-seed attention stability, that is a meaningful effect, not a confound to normalize away: higher-dimensional attention maps are inherently more variable, and our metric is designed to capture, not remove, that effect.
>
> Also, Fig. 4.4.2 shows more than a monotonic stability decline with prompt length: the drop, that is the gap, between successive prompt sets decreases nonlinearly.
>
> We nonetheless tested the reviewer’s suggestion by normalizing Eq. (1) with prompt length raised to the power z:
>
> $$ \text{Prompt length}(p)^z $$
>
> **Stability values for the most unstable (5th) layer in “8-layer, 8-head MLP”:**
>
> | Token Length | Stability (z=0) | Drop % | Stability (z=1/2) | Drop % | Stability (z=1) | Drop % |
> |----------|----------------:|-------:|----------------:|-------:|----------------:|-------:|
> | 5   | 0.802042 |     --     | 0.291460 |      --      | 0.107056 |      --      |
> | 10 | 0.736344 | 8.19% | 0.195075 | 33.07% | 0.051973 | 51.45% |
> | 20 | 0.687822 | 6.59% | 0.131021 | 32.83% | 0.025033 | 51.83% |
> | 30 | 0.667267 | 2.99% | 0.105410 | 19.55% | 0.016681 | 33.36% |
> | 40 | 0.654664 | 1.89% | 0.090621 | 14.03% | 0.012556 | 24.73% |
> | 50 | 0.646651 | 1.22% | 0.080761 | 10.88% | 0.010093 | 19.62% |
>
> \
> **Analysis**: Normalization smooths the curve and lowers absolute stability, but the main trend is unchanged: the most unstable layer still declines with prompt length. It also compresses similarity values and masks variation, making differences clearer in the raw scores. Thus, the curve is more observable using raw scores. Similar analysis for other architectures.
>
> \
> **W2 / Q3 (“The proposed stability metric [...]”), and W3 (“The number of parameters [...]”)**
>
> We thank the reviewer for these questions regarding **model size** and the use of the **attention score** in our paper. Closely related concerns were also raised by **Reviewer 9aBk**, and we address them in greater detail in **W1** and **W2**. Given the limited rebuttal space, we do not duplicate that discussion here and instead respectfully refer the reviewer to our response to **Reviewer 9aBk**.
>
> \
> **Q4 (“Why use cosine similarity [...]”), and Q5 (“[...] optimal transport solver?”)**
>
> We considered distances (Wasserstein, Euclidean) and KL divergence, but chose cosine similarity because it yields normalized, easily comparable scores across seeds, is less sensitive than Wasserstein distance to prompt length and attention entropy, and is simpler to compute and interpret. We do not claim it is the only valid choice, only that it best balances our needs.
>
> \
> **Q6 (“Can two heads in model A [...]”)**
>
> Yes. This is expected and desirable: our method does not assume a bijection between heads across models. As Section 4.3 shows, especially in early layers, redundant heads from a single refit can have very similar attention maps. Similarly, one head in Model A may match multiple heads in Model B. Enforcing bijective alignment, e.g. with Hungarian matching, would therefore be overly restrictive and could worsen matches.
>
> \
> **W4**
>
> We will correct the capitalization in Fig. 2 (line 275).
>
> \
> **W5**
>
> We agree with the feedback. In the main paper, we prioritised the main experimental results and omitted some supporting presentation details (i.e. moved a figure to the appendix). Section 4.4.1 links the L2 norm of the query weight matrices to attention instability, which parallels existing work on L2 norm growth leading to instabilities during training. Based on the feedback, We will clarify Sec. 4.4.1 and move the figure to the main text.
>
> \
> **W6 (“[...] stability of the residual stream [...]”)**
>
> Prior work studied residual-stream similarity in a different context of CKA & similar methods comparisons. Our contribution is twofold: we test it in our refit setting across architectures and ask whether AdamW, as with attention heads, also improves residual-stream stability. Residual streams are much more stable than attention heads across refits, helping explain consistent outputs despite unstable head patterns and suggesting that circuits defined through the residual stream may be more robust than those defined directly through attention patterns. Important takeaways for LLM-circuit researchers.

---

> > ### Author Rebuttal · Reviewer_xxMp · 2026-03-31
> >
> > The authors have addressed my concerns, and I have subsequently increased my score.

---

### Official Review · Reviewer_v2Vu · 2026-03-13

**Soundness:** 2
**Presentation:** 1
**Significance:** 1
**Originality:** 2
**Overall Recommendation:** 2
**Confidence:** 4

**Summary:**

This paper investigates the reproducibility of attention head (described by attention score) in different training repetitions. They repeat training of small Transformers and found that attention heads in the middle layer are more unstable and diverse, and weight decay can promote the training stability of attention heads, etc.

**Compliance With Llm Reviewing Policy:**

Affirmed.

**Final Justification:**

After the author's rebuttal, some of my concerns are addressed. But I still tend to reject this paper since some concerns still remain (e.g., the concerns of attention scores and model size). I would like to increase my score by 0.5.

**Key Questions For Authors:**

Refer to weakness.

**Limitations:**

yes

**Strengths And Weaknesses:**

**Strengths**

- This paper investigates important mechanical interpretability questions, i.e., the repeatability and generalization of circuit/attention heads.

- It is an interesting observation that L2 normalization leads to the stability of the attention head.

**Weakness**

- The conclusion of the paper is scattered. I would rather see a complete picture or model of the training, such as to what extent the training data or certain settings during training have contributed to the emergence of certain circuits or functions. Or, you have confirmed that different layers have different stability/diversity during training, so what deeper reasons are behind this? The lack of such a key framework makes this paper appear as a combination of several scattered experimental reports.

- I don't think using flattened attention scores as the primary representation to characterize the attention head is a reasonable setting. This requires a fairly strong assumption: attention heads with similar attention scores can output similar representations; But attention score is not a good interpretability indicator [1], so I am negative about it. Can you provide some strong evidence or relevant work to prove this?

- Scaling only on models from 7M to 124M does not meet the current scaling requirements of mechanical interpretability. The paper only repeats the seed and training conditions on the minimal LMs (note that the scale is smaller than that of the general mechanical interpretability research objects, which are usually 7B~ LMs). What we would prefer to see is consistent internal representations and circuits across a considerable parameter scale, different model structures (such as using rotation position encoding or position embedding), and on quite inconsistent corpora. I understand that computing resources are scarce, but since you have chosen such a research topic, you should ensure that these resources are available.

- I don't think it makes sense to compare the best match heads of fixed layers. Due to the presence of residual links, some functions or circuits may be delocalized. Therefore, even if the experiment designed in 3.3 is reasonable, the experiment in 4.1 still lacks significance.

[1] Is Attention Explanation? An Introduction to the Debate

---

> ### Author Rebuttal · Authors · 2026-03-31
>
> We thank the reviewer for the thoughtful comments. We found the feedback helpful and have used it to clarify the identified weaknesses (**W**).
>
> \
> **W1: “The conclusion of the paper  [...]”**
>
> Our objective in this work is primarily **measurement and attribution of seed stability**, rather than providing a complete causal account of circuit emergence. As noted in the paper, prior works have examined seed stability from a more functional perspective. However, we believe there is value in comparing attention heads using a metric that is **agnostic to any predefined functional role or circuit interpretation**. This avoids building assumptions about the underlying mechanism of LLMs directly into the comparison metric and reduces the risk of biasing the analysis toward a particular explanatory framework. In addition, any set of hand-selected circuits would necessarily be non-exhaustive.
>
> Even within its present scope, however, the paper already identifies several training- and evaluation-facing drivers that systematically track head stability: (i) computational depth / relative layer position, (ii) optimizer and weight decay, (iii) within-layer uniqueness versus redundancy, (iv) correlates related to norm growth and prompt length, and (v) the relative robustness of the residual stream compared with the attention mechanism. These findings have important implications for the universality and stability of circuits as core mechanistic units of study in LLMs. We do mention these findings in the Introduction to provide a broader context, although we acknowledge that we may not have articulated the overall narrative and implications as clearly as possible.
>
> To address the reviewer’s request for a more complete picture, we will restructure the Conclusion so that the narrative and its implications are stated more explicitly. In particular, we believe the paper offers useful guidance for researchers in mechanistic interpretability and related areas, especially those studying or optimizing the attention block for attention-specific interventions such as steering methods, or distillation objectives based on divergences between attention patterns. More broadly, we believe the contribution of this paper is best understood as a measurement and analysis study of representational stability providing novel insights, rather than as a conventional method paper.
>
> \
> **W2: “I don't think using flattened attention scores [...]”** and **“[1] Is Attention Explanation? An Introduction to the Debate”**
>
> And, **W3: “Scaling only on models [...]”**
>
> We thank the reviewer for these questions regarding **model size** and the use of the **attention score** in our paper. Closely related concerns were also raised by **Reviewer 9aBk**, and we address them in greater detail in **W1** and **W2**. Given the limited rebuttal space, we do not duplicate that discussion here and instead respectfully refer the reviewer to our response to **Reviewer 9aBk**.
>
> \
> **W4: “[...] compare the best match heads of fixed layers [...]”**
>
> We recognise this possibility, and it is precisely why we included Sections 3.3 and 4.3 in the paper to address the same concern raised in W4. Importantly, the results in Section 4.3 exhibit a strongly diagonal-dominant pattern, indicating that most heads are best matched locally, typically within the same layer or at most within a deviation of ±1\pm 1±1 layer.
>
> Based on[1][2], we believe that the location of an attention head within the network is itself an important part of its identity. Even when two heads appear similar, interpreting them as equivalent becomes less straightforward when they occur in very different parts of the model, since later layers operate on substantially different contextual representations than earlier layers. For this reason, it is important to separately consider the case where matching occurs among heads from similar layers, rather than allowing unrestricted delocalization.
>
> This led to a modelling choice: either enforce stronger localisation or allow broader cross-layer matching. Based on the argument above, our empirical observations, and especially the results in Section 4.3, we found that the more localised formulation was the better fit for the broader goals and framing of the paper. Because relaxing this constraint did not materially alter the main results, it does not justify diverting the discussion from the paper’s central findings. We therefore included the cross-layer perspective in the paper, while prioritising the main results and the broader conclusions.
>
> [1] The Remarkable Robustness of LLMs: Stages of Inference? — Lad et al. (2025)\
> [2] How Large Language Models Encode Context Knowledge? A Layer-Wise Probing Study  — Ju et al. (2024)

---

> > ### Author Rebuttal · Reviewer_v2Vu · 2026-04-02
> >
> > Thanks for your reply.
> >
> > W1. W3. The authors promised to make revisions, but I would like to see specific content, such as experimental results or how you would reframe your conclusions. Before the specific content is available, the concerns are still open.
> >
> > W2. I retain my opinion on the attention score. I still consider the similarity (maybe high-level, e.g., kernel alignment) to be a better metric to evaluate.
> >
> > W4. I agree with the explanation.

---

> > > ### Author Response · Authors · 2026-04-04
> > >
> > > We thank the reviewer for the thoughtful further comments and for acknowledging that our earlier clarification partially addressed the concerns. We are happy to clarify the remaining questions below.
> > >
> > > \
> > > **W2**
> > >
> > > We understand and respect the reviewer’s preference regarding the use of attention scores in our paper. Regarding the additional point: “I still consider similarity (perhaps at a high level, e.g., kernel alignment) to be a better metric to evaluate”, we agree that methods such as CKA are important and widely used for comparing representations. However, comparing representations across different spaces is itself nontrivial, since those spaces may differ by transformations that affect similarity measures in subtle ways. As we note in the paper, cross-seed comparisons using methods such as CKA, CCA, and SVCCA are not without limitations and can at times be misleading, as their sensitivity depends on the class of transformations involved, meaning that some differences may go undetected.
> > > By contrast, for cross-seed comparisons in our setting, attention scores provide a more direct object of comparison because they represent token-to-token relationships on a common basis across seeds. We nevertheless agree that kernel alignment may be a more suitable choice in a different setting or for a different objective. Our point is not that one approach is universally better, but rather that the appropriate choice depends on the context and the specific purpose of the comparison.
> > >
> > > \
> > > **W1**
> > >
> > > In response to the reviewer’s request for a more complete picture, we will revise the Conclusion to state the overall narrative and its implications more clearly and explicitly.  We will add the following new section to the Conclusion, weaving these threads together more tightly:
> > >
> > > ```
> > > The observed "stability dip" in middle layers is likely because these intermediate computations operate with the highest degrees of freedom, furthest from the regularizing anchors of the input embeddings and output logits. Without these constraints, individual refits rely on unrestricted parameter expansion, as evidenced by query-weight norm growth, to solve tasks via idiosyncratic, seed-specific attention heads. In practical terms at the single-instance level, middle-layer instability aligns with the structural emergence of an intermediate "compression valley" characterized by high representational divergence from the input space (Skean, 2025), and has been linked to the activation of coherent but input-insensitive semantic concepts that may drive hallucinations (Suresh, 2025).
> > >
> > > By introducing explicit optimization constraints such as coupled weight decay (AdamW), we restrict the parameter space, forcing intermediate layers to converge on more canonical, stable, and universal structures while retaining their task performance. Through an alternate mechanism, the residual stream acts as an architectural constraint, consolidating highly variable head-level outputs into a stable functional pathway. Ultimately, for mechanistic interpretability to scale reliably, circuit extraction must account for this framework, prioritising more highly constrained regions of the network or enforcing regularization during pre-training to provide stronger guarantees for circuit robustness and reliability.
> > >
> > > In particular, we believe the paper offers useful guidance for researchers in mechanistic interpretability and related areas, especially those studying or optimizing the attention block for attention-specific interventions such as steering methods or distillation objectives based on divergences between attention patterns.
> > >
> > > Layer by Layer: Uncovering Hidden Representations in Language Models (Skean et al., 2025)
> > > From Noise to Narrative: Tracing the Origins of Hallucinations in Transformers (Suresh et al., 2025)
> > > ```

---

### Decision · Program_Chairs · 2026-04-30

**Decision:**

Accept (regular)

**Comment:**

Authors ask whether learned attention heads in a transformer are stable across different training runs. They find a number of things, enumerated in the paper abstract. The study has important implications for mechanistic interpretability.

Reviewers mostly found the paper well-executed with a clean methodology and interesting results. Several questions were raised during the rebuttal, including questions about scale of experiments, validity of using attention scores as the representation, and other technical points. Authors provided responses that resolved most of these points.

Overall, three of the four reviewers recommend acceptance and one reviewer votes for rejection. Reviewer v2VU, who voted for rejection, states the reason for his/her low score is due to concerns related to usage of attention scores as the representation and model size. Reviewer 9aBk brought up the same issues, but had a more positive look on the paper and the authors' response. As the AC, my personal view is closer to 9aBk.